

# Causes of the weak emergent constraint on climate sensitivity at the Last Glacial Maximum

Martin Renoult[1], Navjit Sagoo[1], Jiang Zhu[2], and Thorsten Mauritsen[1]

[1]Department of Meteorology, Bolin Centre for Climate Research, Stockholm University, Stockholm, Sweden

[2]Climate and Global Dynamics Laboratory, National Center for Atmospheric Research, Boulder, USA

**Correspondence:** Martin Renoult (martin.renoult@misu.su.se)

**Abstract.** The use of paleoclimates to constrain the equilibrium climate sensitivity (ECS) has seen a growing interest. In particular, the Last Glacial Maximum (LGM) and the mid-Pliocene Warm Period have been used in emergent constraint approaches using simulations from the Paleoclimate Modelling Intercomparison Project (PMIP). Despite lower uncertainties regarding geological proxy data for the LGM in comparison with the Pliocene, the robustness of the emergent constraint
between LGM temperature and ECS is weaker at both global and regional scales. Here, we investigate the climate of the LGM in models through different PMIP generations, and how various factors contribute to the spread of the model ensemble. Certain factors have large impact on an emergent constraint, such as state-dependency in climate feedbacks or model-dependency on ice sheet forcing. Other factors, such as models being out of energetic balance and sea-surface temperature not responding below -1.8°C in polar regions have a limited influence. We quantify some of the contributions and find that they mostly
have extratropical origins. Contrary to what has previously been suggested, from a statistical point of view, the PMIP model generations do not differ substantially. Finally, we show that the lack of high or low ECS models in the ensembles critically limits the strength and reliability of the emergent constraints.

## 1 Introduction

The long-term global mean surface temperature response of the Earth to a doubling of atmospheric $CO_2$ from pre-industrial
conditions, referred as equilibrium climate sensitivity (ECS) is an important metric in constraining future climate change (e.g. Forster et al., 2021; Huusko et al., 2021). However, the estimated range of ECS, particularly its upper bound, has been the subject of debate for more than a century (Arrhenius, 1896). In recent years "emergent constraints"; the building of statistical relationships between two variables of the climate system existing in an ensemble of climate models, allowing to infer one by observing the other, have been extensively used (e.g. Covey et al., 2000; Hall and Qu, 2006). In particular, the possibility of
constraining climate properties that are difficult or impossible to measure or observe, such as ECS, makes emergent constraints a powerful tool. Several paleoclimates have a large forcing and temperature anomaly compared to pre-industrial conditions





and subsequently receive growing interest for such emergent constraint analyses (Crucifix, 2006; Hargreaves et al., 2007; Hargreaves and Annan, 2009; Hargreaves et al., 2012; Schmidt et al., 2014; Hopcroft and Valdes, 2015; Hargreaves and Annan, 2016; Renoult et al., 2020). Other methods have calculated ECS by estimating temperature and radiative forcing from

the proxy record, such as $ECS = \frac{\Delta T_{\text{LGM}}}{\Delta R} \times F_{2\text{xCO}_2}$, where $\Delta T_{LGM}$ refers to the temperature difference between the LGM and pre-industrial state, and $\Delta R$ the difference in radiative forcing, including greenhouse gas forcing, ice sheet forcing and sometimes mineral dust forcing (e.g. Rohling et al., 2012; Sherwood et al., 2020; Tierney et al., 2020). The emergent constraint theory differs from this approach by providing more transparency on the role of global climate models and takes into account the state-dependency as simulated by climate models.

Two paleoclimate events particularly stand out within emergent constraint frameworks: the Last Glacial Maximum (23 - 19 kyrs ago, hereafter LGM) and the mid-Pliocene Warm Period (3.29 - 2.97 million years ago, hereafter Pliocene). The LGM represents peak conditions at the last ice age with a maximum extent of sea ice and ice sheets, minimum greenhouse gas concentrations and high atmospheric loading of dust particles, leading to an estimated radiative forcing of -6.8 Wm$^{-2}$ (-9.6 – -5.2 Wm$^{-2}$, 95% confidence interval (Tierney et al., 2020)). On the contrary, the Pliocene is a warm paleoclimate with a

continental configuration and greenhouse gases concentrations close to modern times, which make the Pliocene a potential analogue of future climates (Dowsett et al., 2009; Haywood et al., 2011). The LGM was one of the initial focus periods of the Paleoclimate Modelling Intercomparison Project (PMIP) Phase 1 (Joussaume and Taylor, 1995) and more than 40 models have simulated the LGM through the four generations of PMIP. The LGM has a relative abundance of proxy data as a result of its proximity to present-day and a large forcing signal and reconstructed LGM temperatures are better constrained than those for

the Pliocene. However, despite the LGM being a more promising candidate for a temperature-based constraint on ECS than the Pliocene, studies using the tropical LGM temperatures have estimated a wider range and a higher upper bound of ECS (0.6 – 5.2 K, 90% interval) than from the Pliocene (0.5 – 4.4 K, 90% interval) (Renoult et al., 2020).

As ECS is defined by global mean temperature, one can argue that in general a model with higher ECS should generate a cooler LGM global temperature than a model with lower ECS. However, previous studies have reported weak correlation

between global LGM temperature and ECS (Crucifix, 2006; Hargreaves et al., 2012), and the more robust constraints were based on tropical LGM temperature (Hargreaves et al., 2012; Schmidt et al., 2014; Hopcroft and Valdes, 2015; Renoult et al., 2020). Using the latter has the advantage of mitigating the large effect of extratropical non-CO$_2$ forcing, namely the Northern hemisphere ice sheets or the Antarctic ice sheets. In addition, the coverage in geological proxy data at the LGM is generally good in the tropics (e.g. Tierney et al., 2020) and until PMIP2, most of the spread in ECS was driven by the spread in tropical

climate feedbacks (Bony et al., 2006; Webb et al., 2006).

Since PMIP3, the strength of the LGM emergent constraint has decreased considerably compared to its Pliocene counterpart. Another disadvantage is that the spread of tropical temperatures within climate models at the LGM is smaller than the spread in global temperatures, owing to the larger amplitude of LGM polar temperatures. For example, Renoult et al. (2020) showed a tropical temperature spread of around 2°C in the whole PMIP ensemble, while Hargreaves et al. (2012) had a spread of more

than 3.5°C in the global temperature of the PMIP2 ensemble. A narrow range is an issue for emergent constraint analysis, as it renders statistical methods more sensitive to outliers and noise. In this study, we define noise as the uncertainty arising from





climate physics in the ensemble of models which impacts the statistical relationship, potentially different from a systematic bias. In Fig. 1, we show that the relationship from the global constraint is nonexistent at the LGM after PMIP3, while the Pliocene constraint can be considered as robust across the model generations. The reasons suggested for a weaker LGM
constraint can be summarized as follow:

– Structural differences in LGM simulations: Despite more models simulating the LGM, Hopcroft and Valdes (2015) suggested that differences in model evolution and in particular the additions of dynamical vegetation and aerosol-related effects were enough to generate discrepancies between PMIP generations. This would affect LGM models more as these span four generations of models, whereas the Pliocene span only the two most recent generations of models. Whilst the
argument of Hopcroft and Valdes (2015) is reasonable, we show in this paper that this explanation alone is insufficient. Notably, models are suspected of being out of equilibrium at the LGM, as well as having various representations of ice sheet forcing, ocean circulation and snow-albedo feedbacks.

– State-dependency between LGM and abrupt4xCO2: Because the LGM is a cold climate, feedbacks may behave differently compared to a warmer climate. This state-dependency usually leads to model-based estimates of ECS from LGM
temperature being lower than from 4xCO2 experiments (Rohling et al., 2012; von der Heydt et al., 2014). We show in this study that several aspects of the climate are affected differently between cooling and warming climates, and could weaken the relationship between the LGM and future climate change. Namely, cloud, albedo and water vapour feedbacks may differ in strength between the LGM and the abrupt4xCO2 state from which the ECS is computed.

The aim of this paper is to provide a framework for the future development of paleo-emergent constraints by addressing the
following question: Why are the LGM regional and global constraints weakly correlated with ECS compared to the Pliocene constraint?

The paper is organized as the following: In Section 2, we describe the PMIP models and the ensemble of analysis performed to investigate the spread of models, as well as summarizing the knowledge on correlations between LGM temperature and ECS. Then, in Section 3, we discuss the differences between tropical and global temperature constraints on ECS. In Section 4,
we show the different aspects of the climate system which can be suspected as significant contributors of noise in the emergent constraints. Finally, we discuss three relevant topics for emergent constraints development: the contribution and amplitude of noise in Section 5, the structure and dissociation of PMIP sub-ensembles in Section 6, and prospects from single-model ensembles in Section 7.

## 2   Methodological consideration

In this section, we summarize how paleo-emergent constraints on ECS have been defined within the literature, as well as discussing the use of surface air temperature (SAT) and sea surface temperature (SST). We describe the PMIP ensembles since PMIP1 and the two models used for feedback analysis and single-model ensembles, MPI-ESM1.2-LR and CESM2.1. We also detail the sampling and resampling methods applied in Section 3 and  6.



## 2.1 Definition of the emergent constraint

The emergent constraint approach in its simplest form is a statistical relationship between two climate variables, where one is predicted and the other an observed predictand. In most cases, the predicted variable is difficult to measure or observe, either because it is an idealised metric such as ECS, or an outcome in the future (e.g. future sea ice change (Boé et al., 2009)). In this paper, the two variables of interest are the temperature of the LGM and the ECS of climate models. In previous studies, temperature and ECS have been interchanged with ECS appearing as both the predicted variable (e.g. Hargreaves et al., 2012;

Schmidt et al., 2014) and the predictand variable (Renoult et al., 2020). Following the definition of emergent constraint as a simple linear relationship, the former can be written as Eq. 1 and the latter as Eq. 2.

$$ECS = \gamma \times T + \delta + \zeta \tag{1}$$

$$T = \alpha \times ECS + \beta + \epsilon \tag{2}$$

Both Eq. 1 and Eq. 2 are defined with slopes $\gamma$ and $\alpha$, and intercepts $\zeta$ and $\beta$, which are obtained by regressing ECS over

temperature, or vice versa. The parameters $\zeta$ and $\epsilon$ usually follow a normal distribution N(0,$\sigma^2$) and represent uncertainty arising in the regression from the spread of the model ensembles. Those parameters $\zeta$ and $\epsilon$ are of particular interest for our study, as they are connected to the aspects of the climate which contribute to the noise of the LGM emergent constraint. It is also possible to add an uncertainty parameter dependent of the predicted variable (i.e., on the left-hand side of Eq. 1 or Eq. 2) and certain statistical methods, such as orthogonal distance regression, take into accounts errors in both predicted and

predictand, and have been used in other emergent constraint analysis (Jiménez-de-la Cuesta and Mauritsen, 2019).

It is debated which statistical method is best applied in emergent constraint frameworks. However, the noise existing in an ensemble of models is independent of the choice of statistical approach used to infer ECS, as models are not built to be related by specific statistical relationships. The Pearson correlation coefficient arising from the relationship between the LGM temperatures and ECS is also independent of the choice of Eq. 1 or Eq. 2 as it is symmetrical. Therefore, discussions regarding

statistical methods is beyond the scope of this study, but we provide ECS estimates when discussing single-model ensembles in Section 7.

Both surface air temperature (SAT) (Hargreaves et al., 2012; Renoult et al., 2020) and sea surface temperature (SST) (Hargreaves and Annan, 2016; Renoult et al., 2020) have been used in emergent constraint studies. From a geological point of view, marine proxies are more abundant than land-based proxies and so using SST is more meaningful. For the LGM, there is a

relatively good coverage of land proxies (Cleator et al., 2020), contrarily to the Pliocene, which gives potential in using either land-only or all-surfaces temperatures. However, ECS values are often computed from SAT in models (e.g. Andrews et al., 2012), which can lead to differences with other temperature variables. For example, MPI-ESM1.2-LR has an ECS reported as 2.77 K in PMIP4 (Kageyama et al., 2021) based on surface temperature, while Mauritsen et al. (2019) showed an ECS of 3.01 K using SAT. SAT is extrapolated and amplified by surface temperature in climate models whereas observations show



the opposite (Gulev et al., 2021). Thus, one could expect emergent constraints using SAT to be inherently biased by this disagreement. In the case of SST, there is little difference between SST and surface temperature for a large part of the globe. In polar regions, discrepancies between the two can be found due to the presence of sea-ice and it is shown later to influence the correlation between polar temperatures and ECS.

There is ambiguity in the definition and calculation of climate sensitivity in climate models. In this paper and unless specif-
ically noted, ECS refers to the methodology of Gregory et al. (2004), an approximation of the long-term equilibrium climate sensitivity from 150-year long perturbed experiment, as it is commonly adopted by the community. However, other studies have used the broader $S$ as "Sensitivity" (Hargreaves et al., 2012; Schmidt et al., 2014; Renoult et al., 2020), and some of the ECS estimates of PMIP1 and PMIP2 models were computed from slab-ocean experiments (Hargreaves et al., 2012; Hopcroft and Valdes, 2015). It is possible that differences in emergent constraints arises from these ambiguities. However, we do not
explore this further.

### 2.2 Variables and models analysed

The climate variables analysed for each model in this study are summarized in Table 1. PMIP spans three decades and the models used to simulate the LGM in PMIP1 and PMIP2 were considerably less complex than more recent models. PMIP1 models were typically Atmospheric General Circulation Models (AGCMs) with low resolution and limited representation
of land surfaces and vegetation. For PMIP2, all models except ECBILTCLIO were Atmosphere-Ocean General Circulation Models (AOGCMs). By PMIP3, a few models started to include complex processes like dynamical vegetation and aerosol-cloud interactions (Hopcroft and Valdes, 2015), while the majority of the models of PMIP4 have implementations of those components.

The availability of the data is based on the current state of each PMIP database, which notably differ from the studies
of Hargreaves et al. (2012), Hargreaves and Annan (2016) or Renoult et al. (2020), as models have been removed or added over time. For PMIP4, which is still ongoing at the time of writing, only SSTs were available to be examined for the majority of the models. We have also included several model variants as they can provide information on the sensitivity of the climate system to specific components. Notably we included: p151 of GISS-E2-R, which has a different ice sheet mask than other PMIP3 models ("Laurentide enhanced"), p2 of MPI-ESM-P which has dynamical vegetation enabled as opposed to the p1,
variants of iLOVECLIM1.1.4 using the ice sheet mask GLAC-1D and of HadCM3B-M2.1aD using the mask GLAC-1D and the PMIP3 mask (blending of ICE-6G, GLAC-1a and ANU), whereas the most commonly used mask is ICE-6G_C within PMIP4 (Kageyama et al., 2021). We exclude the variants from emergent constraint and correlation analyses, similarly to previous studies, but include them in SSTs or effective albedo analyses, as their behaviour can be indicative of structural uncertainties existing in the ensemble.

PMIP1 models were omitted in previous LGM emergent constraint studies. This is due to a number of reasons: PMIP1 models were AGCMs and most of them had prescribed SSTs or ran slab ocean experiments only (Joussaume and Taylor, 1995); their resolutions are low compared to modern standards (e.g. Williamson et al., 1987; Thompson and Pollard, 1997); there are substantial differences in boundary conditions compared to other PMIPs, such as particularly lower ice sheets (Peltier, 1994)





and independent definition of non-$CO_2$ trace gases (Joussaume and Taylor, 1995); the ECS of PMIP1 models and likewise
details of the methods, notably length of integration are difficult to find. The comparison of ECS of PMIP1 models to PMIP2,
PMIP3 and PMIP4 models is therefore challenging. Finally, most of the variables analysed in our study are not available for
PMIP1 models. Thus, we focus our analyses on PMIP2, PMIP3 and PMIP4, but results from PMIP1 are explored in Section 6.

## 2.3    Simulation of LGM climate

### 2.3.1    Partial Radiative Perturbation

We use the coupled Max Planck Institute Earth System Model, version 1.2 at low resolution (MPI-ESM1.2-LR) (Mauritsen
et al., 2019) to investigate aspects of the LGM climate in this study. MPI-ESM1.2-LR contributed to the Climate Modelling
Intercomparison Project Phase 6 (CMIP6) and PMIP4, and its predecessors were present in all generations of PMIP since
PMIP1. MPI-ESM1.2-LR matches the warming observed since pre-industrial well (Mauritsen and Roeckner, 2020), as well as
reconstructions of the LGM SSTs, but is found to be too warm compared to LGM land temperature reconstructions (Kageyama
et al., 2021).

To perform climate feedback analysis, we used an online module of partial radiative perturbation (PRP) in ECHAM6.3. The
method has been described by Wetherald and Manabe (1988) and Colman and McAvaney (1997) and its implementation in
ECHAM was carried out in Meraner et al. (2013). The PRP method exchanges variables of surface albedo, clouds, humidity
and temperature between a stored control state and the current state of interest, and calculate the influence on top-of-atmosphere
(TOA) fluxes arising from each component. In this study, we were interested in exchanging cloud-related properties between
control (LGM and pre-industrial states) and abrupt $CO_2$ doubling and halving experiments, as well as albedo and water vapor
radiative properties, in order to evaluate the strength of the climate feedbacks in the model under conditions different from
pre-industrial.

From pre-industrial conditions, we ran simulations for 150 years with instantaneous and sustained doubling (abrupt2xCO2)
and halving (abrupt0p5xCO2) of $CO_2$ concentrations, following the protocol of Webb et al. (2017). The runs were compared
to a control pre-industrial run of the same length. In the case of our LGM simulation, we ran for 150 years continuing from the
spun up LGM simulations (Marie-Luise Kapsch, pers. comm.) which follow the PMIP4 protocol of Kageyama et al. (2017).
The latter includes changes of ice sheet masks and reduced greenhouse gas concentrations compared to PMIP3. From that
state, we abruptly doubled the LGM $CO_2$ concentration, ran for an additional 150 years and compared it to the control LGM
state to estimate the climate feedbacks.

### 2.3.2    Perturbed physics ensembles

In addition to the PMIP LGM ensemble, we use two single-model ensembles from MPI-ESM1.2-LR and CESM2.1, as well
as an ensemble of the CESM model family. For MPI-ESM1.2-LR, we explore 14 LGM simulations where parameters which
have a large impact on cloud feedbacks and climate sensitivity were perturbed in order to create an ensemble with a range of
ECS values from 2.7 to 4.8 K (Navjit Sagoo, in prep.). Pre-industrial, abrupt4xCO2 and LGM simulations were run for 150





**Table 1.** Summary of the models used in this study and their components. Clim. refers to climatological means. SSTs are reported as tropical (30° S - 30° N) anomalies from pre-industrial. *When compared to other versions of the CESM model family, the ECS of 5.6 K from abrupt2xCO2 in slab ocean mode of Zhu et al. (2021) is used.

| PMIP | Models | Tropical SST | Surface temperature Clim. | Surface temperature Time series | Radiation Clim. | Radiation Time series | AMOC strength | Sea-ice cover | ECS | Reference of ECS |
|---|---|---|---|---|---|---|---|---|---|---|
| PMIP | CCC2.0 | -4.0 | ✓ | | ✓ | | ✓ | | 3.5 | Kattenberg et al. (1996) |
| | CCM1 | -3.70 | ✓ | | ✓ | | ✓ | | 2.1 | Meehl et al. (2000) |
| | CLIMBER2 | -3.19 | ✓ | | ✓ | | ✓ | | 2.63 | Ganopolski et al. (2001) |
| | GEN1 | -1.84 | ✓ | | ✓ | | ✓ | | 2.1 | Thompson and Pollard (1995) |
| | GEN2 | -2.04 | ✓ | | ✓ | | ✓ | | 2.5 | Thompson and Pollard (1997) |
| PMIP1 | GFDL | -1.93 | ✓ | | ✓ | | ✓ | | 3.7 | Kattenberg et al. (1996) |
| | LLN_NH_1 | -3.46 | ✓ | | ✓ | | ✓ | | 1.96 | Gallée et al. (1992) |
| | LMCELMD4 | -0.87 | ✓ | | ✓ | | ✓ | | 3.6 | Treut et al. (1994) |
| | MRI2 | -3.31 | ✓ | | ✓ | | ✓ | | 2 | Cubasch et al. (2001) |
| | UGAMP | N/A | ✓ | | ✓ | | ✓ | | N/A | – |
| | UKMO | -2.84 | ✓ | | ✓ | | ✓ | | 2.7 | Kattenberg et al. (1996) |
| PMIP2 | CCSM3 | -1.73 | ✓ | ✓ | ✓ | ✓ | ✓ | ✓ | 2.12 | Hopcroft and Valdes (2015) |
| | CNRM-CM33 | -1.44 | ✓ | ✓ | ✓ | ✓ | ✓ | ✓ | 2.45 | Hopcroft and Valdes (2015) |
| | ECBILTCLIO | -1.13 | ✓ | ✓ | ✓ | ✓ | ✓ | ✓ | 1.80 | Hopcroft and Valdes (2015) |
| | FGOALS-1.0g | -2.19 | ✓ | ✓ | ✓ | ✓ | ✓ | ✓ | 1.98 | Hopcroft and Valdes (2015) |
| | HadCM3M2 | -2.11 | ✓ | ✓ | ✓ | ✓ | ✓ | ✓ | 3.02 | Hopcroft and Valdes (2015) |
| | IPSL-CM4-V1-MR | -2.33 | ✓ | ✓ | ✓ | ✓ | ✓ | ✓ | 3.80 | Hopcroft and Valdes (2015) |
| | MIROC3.2 | -1.84 | ✓ | ✓ | ✓ | | ✓ | ✓ | 3.73 | Hopcroft and Valdes (2015) |
| | ECHAM53-MPIOM127-LPJ | N/A | ✓ | | ✓ | | ✓ | | 3.58 | Hopcroft and Valdes (2015) |
| PMIP3 | CNRM-CM5 | -1.28 | ✓ | ✓ | ✓ | ✓ | ✓ | ✓ | 3.3 | Kageyama et al. (2021) |
| | GISS-E2-R | -2.39 | ✓ | ✓ | ✓ | ✓ | | ✓ | 2.1 | Kageyama et al. (2021) |
| | IPSL-CM5A-LR | -2.77 | ✓ | ✓ | ✓ | ✓ | | ✓ | 4.1 | Kageyama et al. (2021) |
| | MIROC-ESM | -2.08 | ✓ | ✓ | ✓ | ✓ | | ✓ | 4.7 | Kageyama et al. (2021) |
| | MPI-ESM-P | -2.01 | ✓ | ✓ | ✓ | ✓ | ✓ | ✓ | 3.5 | Kageyama et al. (2021) |
| | MRI-CGCM3 | -2.47 | ✓ | ✓ | ✓ | ✓ | ✓ | ✓ | 2.6 | Kageyama et al. (2021) |
| | CCSM4 | -2.19 | ✓ | ✓ | ✓ | | ✓ | ✓ | 2.9 | Kageyama et al. (2021) |
| | FGOALS-g2 | -2.51 | ✓ | | ✓ | | ✓ | ✓ | 4.4 | Kageyama et al. (2021) |
| | COSMOS | N/A | ✓ | ✓ | ✓ | | | | 4.1 | Kageyama et al. (2021) |
| PMIP4 | AWI-ESM-1-1-LR | -1.66 | ✓ | ✓ | ✓ | ✓ | | ✓ | 3.6 | Kageyama et al. (2021) |
| | MIROC-ES2L | -1.92 | ✓ | ✓ | ✓ | ✓ | | ✓ | 2.7 | Kageyama et al. (2021) |
| | MPI-ESM1.2-LR | -1.56 | ✓ | ✓ | ✓ | ✓ | ✓ | ✓ | 2.77 | Kageyama et al. (2021) |
| | INM-CM4-8 | -1.78 | ✓ | ✓ | ✓ | ✓ | | | 2.1 | Kageyama et al. (2021) |
| | AWI-ESM-2-1-LR | -1.72 | ✓ | ✓ | ✓ | | | | 3.6 | Kageyama et al. (2021) |
| | CESM1.2 | -3.57 | ✓ | ✓ | ✓ | | | | 3.6 | Kageyama et al. (2021) |
| | CESM2.1 | -6.95 | ✓ | ✓ | ✓ | ✓ | ✓ | | 5.15* | Zelinka et al. (2020) |
| | HadCM3B-M2.1aD | -2.66 | ✓ | | ✓ | | | | 2.7 | Kageyama et al. (2021) |
| | IPSLCM5A2 | -2.72 | ✓ | | ✓ | | | | 3.6 | Kageyama et al. (2021) |
| | iLOVECLIM1.1.4 | -1.66 | ✓ | | ✓ | | | | 3.2 | Kageyama et al. (2021) |
| | UofT-CCSM4 | -2.36 | ✓ | | ✓ | | | | 3.2 | Kageyama et al. (2021) |





years or until the simulations crashed. The 150 years of the pre-industrial and abrupt4xCO2 simulations were used to calculate ECS using linear regression (Gregory et al., 2004). The LGM simulations were branched from the equilibrated PMIP4 LGM contribution from the Max Planck Institute, Hamburg (Marie-Luise Kapsch, pers. comm.). The description of the 14 runs is in Table 2. The MPI-ESM1.2-LR single-model ensemble is compared to the single-model ensemble made of perturbed cloud physics versions of CESM2.1 (Zhu et al., 2022), spanning the range of ECS of 3.7 – 6.1 K, calculated using abrupt2xCO2 in slab ocean model configuration.

Additionally, we compare to the 6-member ensemble of different configurations of CESM1.2 and CESM2.1. These coupled simulations have been run to quasi-equilibrium, making this smaller ensemble valuable. This ensemble uses CESM1.2 with CAM5 at ∼2° resolution (Zhu and Poulsen, 2021), CESM1.3 with CAM5 at ∼2° resolution (Zhu et al., 2017), CESM2.1 with CAM6 at ∼1° resolution (Zhu et al., 2021), CESM2.1 with CAM5 at ∼1° resolution (Zhu et al., 2021), CESM2.1 with CAM6 at ∼1° resolution and the CAM5 ice nucleation scheme (Zhu et al., 2022) and CESM2.1 with paleoclimate-calibrated CAM6 at ∼2° (see Zhu et al. (2022) for details).

## 2.4 Resampling and sampling methods

The ensemble size of each phase of PMIP is small with an average of 8 models, in comparison to PlioMIP2 with 16 models. This has been a limitation in studies focused on individual ensembles (Crucifix, 2006; Hargreaves et al., 2012). There is a risk of identifying relationships which are coincidental in smaller ensembles (Caldwell et al., 2014). Therefore, a high level of correlation is required for a constraint to be meaningful in smaller ensembles. For instance, as only four models were available at that time, Crucifix (2006) would have needed a correlation higher than 0.9 from a 95% threshold one-sided test of correlation for a significant relationship between SST and ECS in PMIP2 (Hargreaves et al., 2012). Because of those concerns, resampling and sampling methods are of particular interest, as they can provide new insights on the correlations and emergent constraint relationships.

In this study, we apply one resampling method, the permutation test, and one sampling method, the simple random sampling. For the permutation test, we interchange the sensitivity of PMIP models and generate 10 000 random ensembles to investigate correlation patterns between SST and ECS around the globe, similar to Hargreaves et al. (2012) for the PMIP2 ensemble. This allows us to test whether a pattern is likely to appear by chance, notably as an artifact of small size ensemble. We compute the 5[th] and 95[th] percentiles of the distribution of correlation coefficients of each of the 10 000 permuted ensembles at each grid cell, with the models regridded at 10° resolution to minimize dependency in neighboring cells. If the correlation in the real ensemble is outside of the computed 5 – 95% interval, then such correlation is unlikely to happen by chance. Here, we extend on Hargreaves et al. (2012) as we include the ensemble of PMIP3 and PMIP4 in the permutation tests to check if certain patterns would appear in ensembles of 15 to 26 members. These results are explored in Section 3.

For the case of simple random sampling, we investigate the creation of smaller sub-ensembles of models from the larger PMIP ensemble by randomly sampling models and generating 100 000 smaller PMIP sub-ensembles. The size of the sub-ensemble is set to 8 members, as it is the average size of single-generation PMIP ensembles.

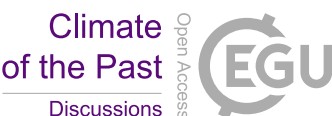

**Table 2.** Summary of the simulations of the single-model ensemble of MPI-ESM1.2-LR. *Unstable runs, where the temperature is an estimate of the last 50 years before numerical crash. **Iris effect implementation of Mauritsen and Stevens (2015). †JM19 refers to runs with all the changes of the table, as well as an increase of the relative humidity threshold for cloud formation at high model level, decrease of entrainment rate in shallow convection and decrease of minimum excess buoyancy, as described by Jiménez-de-la Cuesta and Mauritsen (2019).

| No. | Change | Standard value | Changed value | ECS | LGM temperature change (°C) |
|---|---|---|---|---|---|
| 1 | No change (Standard LGM) | | | 2.92 | -3.86 |
| 2 | With iris effect** | | | 2.65 | -3.70 |
| 3 | Relative humidity threshold for cloud formation in the lowest model level | 0.973 | 0.988 | 3.06 | -3.83 |
| 4 | Determination of vertical profile of the relative humidity threshold for cloud formation between near-surface and upper troposphere | 2 | 1 | 3.28 | -3.91 |
| 5 | Value for liquid-cloud inhomogeneity factor without convection or with deep/mid-level convection | 0.8 | 0.74 | 2.94 | -3.89 |
| 6 | Conversion factor of cloud water to precipitation | $2.5e^{-4}$ s$^{-1}$ | $7.5e^{-4}$ s$^{-1}$ | 2.87 | -3.83 |
| 7 | Gravity wave drag coefficient | 0.2 | 0.5 | 2.92 | -3.86 |
| 8 | Threshold for separation between cloud liquid water and cloud ice; larger values mean more liquid water | $5e^{-6}$ kg.m$^{-3}$ | $1.5e^{-5}$ kg.m$^{-3}$ | 2.87 | -4.28 |
| 9 | JM19† with modified cloud liquid water and ice separation threshold | $5e^{-6}$ kg.m$^{-3}$ | $4e^{-4}$ kg.m$^{-3}$ | 3.96 | -5.22 |
| 10 | Entrainment rate for shallow convection | $3e^{-3}$ m$^{-1}$ | $3e^{-4}$ m$^{-1}$ | 3.73 | -4.34 |
| 11* | Threshold for separation between cloud liquid water and cloud ice; larger values mean more liquid water | $5e^{-6}$ kg.m$^{-3}$ | $2.5e^{-5}$ kg.m$^{-3}$ | 3.54 | -4.27 |
| 12* | Threshold for separation between cloud liquid water and cloud ice; larger values mean more liquid water | $5e^{-6}$ kg.m$^{-3}$ | $5e^{-5}$ kg.m$^{-3}$ | 3.72 | -5.13 |
| 13* | JM19† | | | 4.74 | -4.87 |
| 14* | JM19† with iris effect** | | | 4.77 | -5.22 |



## 3 Regional correlations

The correlation between SST and ECS at the LGM has important regional and generational disparities. A negative correlation between SST and ECS for the LGM is expected, as it implies models with high ECS would simulate a larger cooling as opposed to models with low ECS (Hargreaves et al., 2012). However, patterns of weak, near-zero or positive correlations can be seen around the globe in most ensembles. This is opposed to the correlation between ECS and SST in abrupt4xCO2 simulations, where the correlation is almost globally positively significant (not shown).

We summarize the correlation between SST and ECS at the LGM and the Pliocene among different PMIP generations in Fig. 2. Significance is calculated from a one-sided t-test at 95% threshold. Correlation maps assume the temperature of cells to be strictly independent of neighbouring cells, which is an approximation of reality. Nevertheless, they provide a useful qualitative representations of the sources of noise in the emergent constraint between SST and ECS.

For the Pliocene, the correlation is significant in the tropics during PMIP3, and extends far into the extratropics during 230 PMIP4. Areas of low or negative correlation are the Southern Ocean and the North Atlantic. These patterns of correlation are close from the ones arising from ECS and SST in abrupt4xCO2 experiments. From these two model generations, emergent constraints between SST and ECS seem robust for the Pliocene.

The evolution of LGM-based emergent constraints is less clear across the generations. In PMIP2, there is a significant negative correlation in the tropics, as expected when correlating cooling temperatures to increasing ECS, and positive correlation in 235 the Southern Ocean. A regional positive correlation means that more sensitive models cool less in those regions in their LGM simulations than low climate sensitivity models. In the PMIP3 ensemble, the patterns are broadly split equally between positive and negative correlation, but remain weak and below significance on the correlation map. In PMIP4, the correlation is negative and highly significant in most of the globe when CESM2.1 is included. This is caused by the high ECS and resulting large cooling of CESM2.1 which strengthens the constraint. If CESM2.1 is filtered out, the correlation drops and is insignificant in 240 most parts of the globe.

Interestingly, the Northern Atlantic ocean SST exhibits a relatively large positive correlation with ECS in LGM simulations. Whether or not CESM2.1 is included in the ensemble, there are significant tropical correlations. However, the tropical patterns have a low correlation (minimum of -0.3) when CESM2.1 is not included, and the global correlation is close to zero, whereas tropical patterns have a high correlation (close to -0.6) when CESM2.1 is in the ensemble. One could reason that if the 245 robustness of an emergent constraint is based solely on the presence of a single model, the constraint itself may not be reliable, or such a single model needs to be considered separately of the ensemble. The value of very low or high ECS models, like CESM2.1, is discussed further in Section 6.

The presence of near-zero or positive correlations in the Southern Ocean at the LGM is particularly interesting and is seen in most ensembles. The phenomenon was observed in the PMIP2 ensemble (Hargreaves et al., 2012) and is visible in the 250 PMIP3 ensemble and the combination of PMIP2 + PMIP3 (Fig. 2). This unexpected correlation is not isolated to the LGM, as it has been observed to a lesser extent during the Pliocene (Fig. 2 and Hargreaves and Annan (2016)) and abrupt4xCO2





simulations. Hargreaves et al. (2012) suspected the positive correlation to arises from the small size of the PMIP2 ensemble, but its existence in larger ensembles contradicts that hypothesis.

In Fig. 3, we show that from permuted individual PMIP2 and PMIP3 ensembles, the Southern Ocean positive correlation is
likely to appear by chance in the real individual PMIP2 and PMIP3 ensembles. However, the combination PMIP2 + PMIP3 leads to a large part of the Southern Ocean positive correlation passing the statistical significance test, indicating that it is unlikely to have such positive correlation pattern appearing by chance within this 15-model ensemble. Curiously, the Southern Ocean true correlation in PMIP2 + PMIP3 + PMIP4 falls within the permuted ensemble interval, raising the question if such pattern is influenced by PMIP4 models. When CESM2.1 is included, the highly negative tropical correlation of the true PMIP
ensemble passes the statistical significance test, indicating it is unlikely to appear by chance. When CESM2.1 is removed, only the tropical Indian ocean passes the significance test (not shown).

Based on the above analysis, the robustness of a relationship between ECS and LGM SST is compromised. We shall argue next that this is caused by numerous sources of noise acting on the relationship between LGM global cooling and ECS. Moreover, the differences in correlation between the extratropics and the tropics may arise from the sources of noise which are
essentially extratropical-based, which reinforces the use of tropical LGM SST over global SST for emergent constraints.

## 4   Investigation of LGM climate physics

In this section, we describe and analyse several potential sources of noise and biases which may impact the emergent constraint between LGM temperatures and ECS. This assessment targets all climate components, i.e. the atmosphere, ocean, land surface, but also investigate whether potential biases preferentially affect models individually, through PMIP generations, or the
ensemble as a whole. The contribution of each source to the uncertainty of the emergent constraint is given in Section 5.

### 4.1   Temperature drift and energy leakage

Climate models which have not been spun up sufficiently, i.e. have not been run for the time required for a model to reach its steady state, may experience drift of their climate state. This was shown by Mauritsen et al. (2012) on pre-industrial simulations in the CMIP3 and CMIP5 ensembles where some model pre-industrial simulations would drift as far as 1°C from their initial
temperature within 500 years. Ideally, when in a steady state, climate models would also have a TOA radiation balance at equilibrium, implying that energy is neither created nor lost artificially.

We show the time evolution of surface temperature and TOA imbalance in models simulating the LGM and pre-industrial climates in Fig. 4, and report the drifts of temperature per century in Table 3. As limited computational power was available at the time, PMIP2 models could be suspected to be further out of equilibrium than newer model generations. However,
several of them applied acceleration techniques to reach near-equilibrium state, namely forced adjustment of SSTs to glacial SSTs (Haney, 1971; Hewitt et al., 2003) or acceleration of abyssal temperatures (Bryan, 1984; Shin et al., 2003), though for the most parts, details of the spin up procedures are usually undocumented. In PMIP2, the largest drifts are for FGOALS-1.0g and MIROC3.2, respectively of -0.116°C.century$^{-1}$ and -0.050°C.century$^{-1}$ (Braconnot et al., 2007). In PMIP3 and PMIP4, most





drifts are comprised between -0.1 and -0.05°C.century$^{-1}$, with two models of PMIP3 standing out: MIROC-ESM and MRI-
CGCM3, with drifts of 0.23°C.century$^{-1}$ and 0.19°C.century$^{-1}$, respectively. This could be connected to the abandonment
of acceleration techniques when modelling centres could afford running the ocean models to near-equilibrium. As opposed to
FGOALS-1.0g and MIROC3.2, the drifts of MIROC-ESM amd MRI-CGCM3 are positive and would indicate a warm-drifting
LGM equilibrium temperature, implying that the LGM temperature estimate is low-biased in those models.

The models CCSM4 and IPSL-CM5A-LR appear to be either far from equilibrium or have substantial leaks and gains of
energy, respectively, compared to their pre-industrial states which lie near zero radiation balance (Fig. 4). This could imply
that energy conservation in these models is state-dependent, and that their simulated LGM cooling is biased by model artifacts
acting differently at pre-industrial. All in all, we cannot identify a systematic bias of the PMIP models simulating the LGM
regarding their drift or state-dependent energy conservation. Although there are fewer models with a gain than a loss of energy,
there is a wide range of TOA energy imbalances as well as temperature drifts. In particular, the hypothesis that PMIP2 models
would be either more out-of-balance or drifting more owing to computation limitations does not hold when compared with
more recent models.

## 4.2  SST freezing temperature

Paleo-emergent constraints often rely on SSTs as the observable. However, in a cooling climate such as the LGM this can
be problematic as SST can not go below the average freezing point of -1.8°C, which would lead to a decoupling between
ECS and SST. We plot polar (70° N and northwards, 70° S and southwards) SSTs for PMIP2, PMIP3 and PMIP4 models for
both pre-industrial and LGM simulations in Fig. 5, and examine whether models with cold-biased pre-industrial SSTs or high
climate sensitivities exhibit physically bounded SST under the LGM forcing.

The model with the highest ECS is CESM2.1 at 5.15 K (Zelinka et al., 2020), and simulates a global surface temperature
cooling at the LGM of -11.3°C (Zhu et al., 2021). However, its south polar (average of 70° S - 90° S) LGM SST is -1.99°C,
and its pre-industrial SST -1.66°C. The temperature difference of -0.35°C clearly indicates that the Southern Ocean LGM
cooling in CESM2.1 is limited by the lower bound on SST, resulting in a decoupling between its high ECS and low simulated
temperature anomaly.

Out of 32 models, 22 have a polar cap with mean SSTs close to the -1.8°C physical bound in either one or both hemispheres
in their LGM simulations. If the pre-industrial SST is close to -1.8°C, this will result in a minimal LGM temperature anomaly.
8 models are affected, but only FGOALS-1.0g acts as such in the Arctic ocean. As for models reaching the physical bound
owing to their LGM cooling, 17 models display such behaviour in the Arctic ocean, but only 6 models in the Southern ocean.
There is no clear disparity among generations: models with cold pre-indusrial SST are found in all PMIP, as well as models
with large LGM cooling.

The analysis is naturally sensitive to the chosen latitudes. When extending the regions to instead being poleward of 60°
N and 60° S, only CESM2.1 and MIROC-ESM reach the freezing threshold in the Arctic due to LGM cooling, as well as
FGOALS-1.0g due to its extensively cold pre-industrial SSTs (not shown). This is misleading and shows that regional bias on
SST, such as the one poleward of 70° N and 70° S, may be hidden within global SST. It is unclear why large LGM cooling




**Table 3.** Temperature trends in °C.century$^{-1}$ for models of PMIP2, PMIP3 and PMIP4. For PMIP2, we report the results of Braconnot et al. (2007). Trends in PMIP3 and PMIP4 models are computed as the difference between the mean of the last 30 years and the mean of the first 30 years, normalised by simulation length. *Simulations have a minimum length of 100 years, but may be up to 200 years long (Braconnot et al., 2007).

| PMIP | Model | Length (years) | LGM trend (°C.century$^{-1}$) | Pre-industrial trend (°C.century$^{-1}$) |
|---|---|---|---|---|
| PMIP2 | CCSM3 | 100* | -0.010 | -0.012 |
| | ECBILTCLIO | 100* | -0.009 | -0.025 |
| | FGOALS-1.0g | 100* | -0.116 | -0.025 |
| | HadCM3M2 | 100* | 0.032 | $-3\times10^{-4}$ |
| | IPSL-CM4-V1-MR | 100* | -0.039 | 0.019 |
| | MIROC3.2 | 100* | -0.050 | $6\times10^{-4}$ |
| PMIP3 | CNRM-CM5 | 200 | -0.090 | 0.015 |
| | GISS-E2-R-p150 | 100 | -0.013 | 0.021 |
| | IPSL-CM5A-LR | 200 | -0.072 | 0.011 |
| | MIROC-ESM | 100 | 0.232 | 0.111 |
| | MPI-ESM-P-p1 | 100 | -0.080 | -0.002 |
| | MRI-CGCM3 | 100 | 0.186 | 0.024 |
| | CCSM4 | 100 | Climatology | -0.019 |
| PMIP4 | INM-CM4-8 | 200 | -0.069 | -0.026 |
| | MIROC-ES2L | 100 | -0.021 | -0.006 |
| | MPI-ESM1.2-LR | 500 | 0.006 | -0.005 |
| | AWI-ESM-1-1-LR | 100 | -0.099 | -0.014 |
| | CESM2.1 | 150 | -0.103 | 0.031 |

and cold pre-industrials SST are preferentially found in the Arctic and Antarctic oceans, respectively. It may be connected to how heat is transported by the ocean circulation northward. In the following sections, we show that there are large disparities
in representing the LGM ocean circulation within PMIP models.

### 4.3 Ice sheet forcing

The Laurentide and Fennoscandian ice sheets covered large parts of Northern hemisphere continents and were two main contributors of the negative forcing during the LGM (e.g. Braconnot et al., 2012). Whereas their geographical extent are reasonably



well-constrained (Clark and Mix, 2002; Svendsen et al., 2004; Kleman et al., 2013), their topography and volume remain a
challenge to determine as proxy records only provide limited information. Through PMIP generations, the altitude and reso-
lution of the ice sheet masks have been considerably modified, but the forcing assessments accounting for such modifications
are scarce (Abe-Ouchi et al., 2015; Zhu and Poulsen, 2021).

In Fig. 6, we show that high variance in outgoing surface shortwave radiation is found either on or around the Laurentide
and Fennoscandian ice sheets in the different generations of PMIP. Likewise, the efficacy of LGM ice sheet forcing, i.e.
the contribution of ice sheets to temperature change with respect to a doubling of atmospheric $CO_2$, is found to be model-
dependent (Shakun, 2017; Zhu and Poulsen, 2021). If the temperature change induced by the ice sheets can be written as
$\Delta T_{IS} = \epsilon \times \frac{F_{IS}}{-\lambda}$ and the LGM temperature anomaly as $\Delta T = \Delta T_{IS} + \Delta T_{other}$, with $\epsilon$ the ice sheet forcing efficacy, $F_{IS}$ the
forcing from ice sheets only and $\lambda$ the global climate feedbacks, then the contribution of ice sheets to global LGM cooling is
Eq. 3.

$$\frac{\Delta T_{IS}}{\Delta T} = \frac{\epsilon \frac{F_{IS}}{-\lambda}}{\epsilon \frac{F_{IS}}{-\lambda} + \frac{F_{other}}{-\lambda}} = \frac{\epsilon F_{IS}}{\epsilon F_{IS} + F_{other}} \tag{3}$$

The ratio $\frac{\Delta T_{IS}}{\Delta T}$ approximately varies between 0.2 and 0.7 in 12 model simulations (Shakun, 2017). Orbital forcing is expected
to be small (e.g. Liu et al., 2014), thus we set $F_{other}$ to the well-constrained forcing coming from greenhouse gases at the LGM
of $F_{GHG} = -2.8$ Wm$^{-2}$ (Köhler et al., 2010). For $F_{IS}$, we test values with a range of -1.8 Wm$^{-2}$ to -5.2 Wm$^{-2}$ (Braconnot
et al., 2012; Tierney et al., 2020). If ice sheets contribute to 20% of temperature change, the efficacy of the forcing is between
1.9 for the low ice sheet forcing and 0.7 for the high ice sheet forcing. For a contribution of 70%, the efficacy is between 3.6
and 1.3.

In CESM1.2 and CESM2.1, the ice sheet forcing efficacy is 1.1 and 1.9, respectively (Zhu and Poulsen, 2021; Zhu et al.,
2021), which are mostly connected to the cooling of northern hemisphere SSTs in connection with changes in wind patterns
due to the topography of the ice sheets. The ice sheets, and notably the Laurentide, are known to disturb atmospheric and ocean
circulation. Notably, it impacts Arctic region temperature (Liakka and Löfverström, 2018), Atlantic ocean surface winds and
deep water formation (e.g. Muglia and Schmittner, 2015; Sherriff-Tadano et al., 2018), and local cloud feedbacks (Zhu and
Poulsen, 2021).

In summary, the variance in reflected shortwave radiation due to ice sheets topography is likely to generate noise, as the
efficacy of the forcing is found to be model-dependent. Current models do not indicate if the ice sheet forcing efficacy is above
or below unity, but show a substantial inter-model spread. Since the ice sheet forcing is roughly half the total forcing in LGM,
this means that the ice sheet forcing, and its efficacy, could be a major source of noise in the model relationship between LGM
cooling and ECS.

### 4.4  Surface albedo feedbacks

In response to a cooling climate, the surface albedo feedback ($\lambda_a$) is thought to strengthen as sea-ice, snow cover and more
reflective vegetation biomes extend. Whereas there is a broad agreement among models on this amplification, the amplitude of





**Table 4.** Global and tropical (30° S - 30° N) surface albedo feedback ($\lambda_a$) values computed with PRP method for three simulations performed with MPI-ESM1.2-LR.

|  | Global $\lambda_a$ | Tropical $\lambda_a$ |
|---|---|---|
| abrupt2xCO2 | 0.29 | -0.03 |
| LGM abrupt2xCO2 | 0.34 | 0.02 |
| abrupt0p5xCO2 | 0.55 | 0.12 |

the state-dependency in $\lambda_a$ is likely model-dependent. In Table 4, we report the tropical and global $\lambda_a$ after abruptly doubling $CO_2$ from pre-industrial and LGM conditions, and abruptly halving $CO_2$ from pre-industrial conditions in MPI-ESM1.2-LR.

The global value of $\lambda_a$ is 0.21 Wm$^{-2}$ higher when halving $CO_2$ compared to the LGM experiment in MPI-ESM1.2-LR, which itself is only slightly higher than in the abrupt2xCO2 experiment. In the tropical area, both abrupt2xCO2 simulations from pre-industrial or LGM conditions show almost the same $\lambda_a$ value close to zero, while the abrupt0p5xCO2 $\lambda_a$ is 0.1 Wm$^{-2}$ larger. Similar findings have been made by Colman and McAvaney (2009), Yoshimori et al. (2009) and Zhu and Poulsen (2021).

The strength of the state-dependency on $\lambda_a$ remains difficult to estimate, but the inter-model spread in model abrupt4xCO2 simulations is substantial, with a standard deviation of 0.09 Wm$^{-2}$K$^{-1}$ (Zelinka et al., 2020), which could be used as an indicator of the spread in state-dependency. This magnitude would not be inconsistent with the available anecdotal single-model evidences.

In the following sections, we explore the individual contributions of snow, vegetation and sea-ice albedo feedbacks at the LGM, and how their inter-model differences may act as sources of noise.

### 4.4.1 Snow and vegetation albedo feedbacks

Hopcroft and Valdes (2015) hypothesized that snow and vegetation albedo feedbacks might play a part in weakening the robustness of the emergent constraint by generating noise within the ensemble. We show in Fig. 7 the maps of effective surface albedo of MIROC-ESM, MRI-CGCM3, GISS-E2-R-p150 and p151 and CNRM-CM5, as well as a comparison with the PMIP3 ensemble mean. Those models were characterized with unusual radiation balance changes over land compared to PMIP2 and PMIP3 models, which was suspected of generating noise in the PMIP3 emergent constraint analysis (Hopcroft and Valdes, 2015).

The model CNRM-CM5 particularly stands out as it has patches of unusually low albedo on top of the Laurentide, Greenland, Fennoscandian and Antarctic ice sheets. Climate models usually display different albedo for bare ice and snow, and the snow albedo is often dependent on various factors, such as snow thickness, temperature and sometimes the history of the conditions the snow has experienced. A comparison with the simulated LGM snow cover reveals that the parts of the Laurentide, Greenland, Antarctic and Fennoscandian ice sheets, as well as the Andes and Himalayan mountain range, which show high




effective surface albedo, are connected with relatively important snow cover. On the contrary, the patches of low albedo are connected with less snow cover, or no snow cover at all in the case of Antarctica. Central to northern Asia is also slightly covered by snow, and shows also a relatively low albedo. This could indicate that the snow-free albedo of land in CNRM-CM5 is low, in particular compared to the albedo of snow-covered areas. It is not necessarily singular, as the albedo of vegetation is low and there is a large range in bare ice sheet albedo (Cuffey and Paterson, 2010), but it contrasts with the other models, where

either the land ice albedo is close to the albedo of fresh snow of 0.8, or the ice sheets and forests are entirely covered by snow. Regarding the Laurentide, Antarctic and Greenland ice sheets, the areas of high snow cover are connected to lower topography, and restricted snow cover usually happens in areas of topography higher than 4000 meters (Abe-Ouchi et al., 2015). Thus, it may be that topography limits more snow fall in CNRM-CM5 than other models, which leads the average surface albedo to be dominated by the albedo of bare ice sheet.

GISS-E2-R has a high effective albedo in Northern Asia, leading to relatively cold local LGM temperatures for its low ECS. GISS-E2-R uses the prescribed LGM vegetation of Ray and Adams (2001) rather than the prescribed pre-industrial or dynamical vegetation as suggested for the PMIP3 experimental design. However, Hopcroft and Valdes (2015) noted that models with dynamical vegetation which simulated a greater loss of forest cover (e.g. MIROC-ESM) were not as cold as GISS-E2-R, implying that the vegetation map is not fully responsible for the behaviour of GISS-E2-R. Instead, snow albedo and how the

model handles its interaction with vegetation is likely to be responsible for this higher albedo. The representation of snow and canopy in high latitude forests was considered as highly challenging for GCMs by the Snow Model Intercomparison Project (SnowMIP) Phase 2 (Essery et al., 2009). In the case of GISS-E2-R, the albedo of the canopy is prescribed based on vegetation type, but the reflectivity of the canopy increases when snow sticks to it (Qu and Hall, 2007; Thackeray et al., 2018). It is likely that a fraction of snow remains over the canopy in GISS-E2-R, whether it is due to too much snow, properties of the vegetation,

or a fixed process simulating a high albedo for every snowfall over vegetation.

Vegetation processes and feedbacks are also important, as the implementation of dynamical vegetation has been suspected of contributing to the spread in the PMIP3 ensemble (Hopcroft and Valdes, 2015). In PMIP3, only MIROC-ESM and MPI-ESM-P-p2 included dynamical vegetation. MIROC-ESM is characterized by a substantial decrease of surface albedo in the Sahara, due to its dynamical vegetation response under LGM forcing, leading to a weak total surface albedo feedback. In

Fig. 7, the comparison between MPI-ESM-P-p1 and MPI-ESM-P-p2 shows an increased effective surface albedo over Siberia compared to the static vegetation version which is likely induced by the replacement of trees with lower vegetation, snow-covered areas. MRI-CGCM3 is a special case as it does not have dynamical vegetation but it exhibits a relatively strong albedo feedback, which is induced by the albedo of the areas of the LGM land mask due to the lower sea-level set to the albedo of bare soil (Hopcroft and Valdes, 2015).

It is reasonable to argue that the implementation of processes such as dynamical vegetation in PMIP3, and some aspects of snow-albedo feedbacks might play a role in causing spread in the ensemble, as hypothesized by Hopcroft and Valdes (2015). However, they have a limited geographical impact and only concern a few models. Therefore, it is difficult to show that the lack of robustness of the emergent constraint would only be induced by these factors. We note that both GISS-E2-R and CNRM-



CM5 have seasonal cycles of snow-albedo feedbacks matching observations of modern times (Fletcher et al., 2015; Thackeray
et al., 2018), despite being locally either too cold (GISS-E2-R) or too warm (CNRM-CM5) at the LGM.

### 4.4.2    Sea-ice albedo feedback

Owing to the importance of sea-ice extent at the LGM, it is plausible that the LGM sea-ice albedo feedback may also contribute
to a decoupling between ECS and surface temperature or SAT in those regions. In Fig. 6, we show that there is a high variance
in surface shortwave radiation in polar oceans. This high variance is located towards the sea-ice edge, as there are disparities in
sea-ice extent among models, with implications on the sea-ice albedo feedback. For instance in Fig. 7, the model GISS-E2-R
has a small surface albedo in the Southern Ocean, indicative of a limited sea-ice extent, whereas the PMIP3 ensemble mean
shows an extent equatorwards of 60° S.

A spread in sea-ice albedo feedback is not necessarily an issue for emergent constraint, similarly to snow and vegetation
albedo feedbacks, but it becomes a concern if models show a behaviour in sea-ice albedo feedback which is not expected at
first from their ECS value. This is the case for FGOALS-1.0g and ECBILTCLIO, two models with an ECS value below 2 K,
which show large extent of sea-ice, where FGOALS-1.0g Arctic ocean sea-ice extent reaches 40° N at the LGM, compared to
55° N at pre-industrial (not shown). The strength of the sea-ice albedo feedback is also influenced by the presence or absence
of snow, whereas the albedo of snow-free sea-ice varies greatly among models. Therefore, we contend that sea-ice albedo
feedback might be a contributor of noise within the ensemble of PMIP, arising from a few models. Similarly to the snow and
vegetation feedbacks, the contribution is regional, and therefore the sea-ice albedo feedback is unlikely to be the main driver
for the weakness of the emergent constraint.

### 4.5    Ocean structure and dynamics

Accurate representation of ocean circulation in models is necessary as it impacts heat transport and SST. The Atlantic merid-
ional overturning circulation (AMOC) and Southern ocean dynamics are important regulators of the climate via many roles
which include energy and heat transport, deep water formation and interactions with sea-ice. In this section we investigate the
contribution of noise in the emergent constraint framework from the AMOC and the Southern ocean via the representation of
two water masses, North Atlantic deep water (NADW) and Antarctic bottom water (AABW).

### 4.5.1    The AMOC

Both CMIP5 and CMIP6 model ensemble means show a decline in AMOC strength in response to future warming scenar-
ios (Weijer et al., 2020; Lee et al., 2021). This results in a northern hemisphere cooling as less heat is transported from the
equator to the Arctic (Jackson et al., 2015). Geological reconstructions of the LGM indicate that the glacial AMOC was as
vigorous as present day (Yu et al., 1996; Lynch-Stieglitz, 2017), 30 to 40 percent weaker (McManus et al., 2004) or potentially
stronger (Lippold et al., 2012). Proxy data do agree that during the LGM NADW shoaled and there was a northwards intrusion
of AABW due to increased sea-ice extent in the Southern ocean (Rahmstorf, 2002; Lynch-Stieglitz et al., 2007; Böhm et al.,





2015; Lippold et al., 2016). Recently, Kageyama et al. (2021) showed that most of the PMIP4 LGM simulations have substantial intrusion of AABW, and a shoaling of NADW for two of the models, but earlier AMOC simulations in PMIP3 and PMIP2 models were more divided (e.g. Otto-Bliesner et al., 2007; Sherriff-Tadano and Klockmann, 2021).

The representations of the AMOC can be classified in three categories, with examples provided in Fig. 8. For example, CCSM3 is one of the models which matches the proxy data fairly well, with a substantial northwards intrusion of AABW
and a shoaling of NADW. MPI-ESM1.2-LR is an example of a model simulating little changes between the LGM and the pre-industrial. Whereas it agrees with proxy indicative of the strength of the AMOC being similar to pre-industrial, these models do not show a substantial intrusion of AABW as seen in the proxy record, and in most cases the NADW does not shoal substantially. Finally, CNRM-CM5 is an example of a model which strongly disagrees with proxy data. Here, the NADW reaches the seafloor at the LGM, and although glacial AMOC can be stronger than pre-industrial, proxy and 2D models do not
support the LGM AMOC to be more than 10 Sv stronger than during pre-industrial (Lippold et al., 2012), which is the case for CNRM-CM5 and other models (Otto-Bliesner et al., 2007; Muglia and Schmittner, 2015).

It is unclear if an inaccurate representation of the AMOC has an influence on global or hemispheric temperatures, and consequently the emergent constraint between SST and ECS. Indeed, we find no relationship between by how much a model matches the proxy reconstruction of the AMOC and its northern hemisphere surface temperature. Models which simulate a
substantial strengthening of the AMOC under the LGM forcing are not necessarily warmer, which could have been expected based on the pre-industrial simulations of Jackson et al. (2015). Moreover, the reasons behind the spread in modelled AMOC structure and strength are unclear (Sherriff-Tadano and Klockmann, 2021). Notably, the effects of density (Weber et al., 2007), salinity (Otto-Bliesner et al., 2007), wind changes driven by the Laurentide topography (e.g. Muglia and Schmittner, 2015; Sherriff-Tadano et al., 2018), or limited Antarctic sea-ice formation (Marzocchi and Jansen, 2017) have been suspected of
influencing the AMOC in PMIP models.

Recently, Marzocchi and Jansen (2017) suggested that models behaving similar to CNRM-CM5 might simulate a substantial deepening of NADW due to too short spin-up periods. In an extended 900-year long run from its spin-up phase, the AMOC of CCSM4 shallowed by 400 meters and weakened by 9 Sv, leading to values closer to proxy data (Rahmstorf, 2002; Böhm et al., 2015; Lippold et al., 2016). While the drift does not slow down after 900 years, it is not visible within the 100 year
time series of PMIP3 as it is obscured by natural variability. We perform a similar analysis with MPI-ESM1.2-LR, a model which shows little change in AMOC strength and position between the LGM and pre-industrial. We run 621 years from the spin-up phase, which lasted 3850 years (Kapsch, pers. comm.). The trend in AMOC strength is lower than 0.05 Sv.century$^{-1}$ compared to almost 1 Sv.century$^{-1}$ for CCSM4, thus we conclude that there is no substantial drift of AMOC in MPI-ESM1.2-LR. This could indicate that the observations of Marzocchi and Jansen (2017) are model-dependent and only affect CCSM4,
or that recent PMIP4 models have an AMOC closer to equilibrium owing to their longer integration times (i.e. 6760 years for MIROC-ES2L, Ohgaito et al. (2021)).

It is likely that the inter-model disagreements on AMOC representations contribute to the spread of models in the emergent constraint framework. Nevertheless, we do not find a clear relationship between AMOC representations and simulated LGM





temperatures, which indicate that either the different AMOC representations are compensated on a broader scale, or they are
limited sources of noise for the emergent constraint relationship.

### 4.5.2   Southern ocean dynamics

Geological reconstructions strongly indicate that AABW intruded further north in the LGM owing to Southern ocean dynamics.
This intrusion is seen in very few models until PMIP4 (Kageyama et al., 2021). The models which best match proxy records
show extensive annual Southern ocean sea-ice, which is known to play a part in the ocean circulation via brine rejection (Mar-
zocchi and Jansen, 2017). Recently, Zhu and Poulsen (2021) showed that the strong coupling existing between Southern ocean
convection, upper ocean heat convergence and sea-ice contributes to a stronger LGM cooling of ∼1°C in CESM1.2. Following
that, we hypothesize that models with a large intrusion of AABW, thus highly convective Southern ocean, could be relatively
colder than other models with respect to their ECS. Similar to the AMOC, we do not find a clear relationship between Southern
hemisphere temperature and the representations of the Southern ocean well-matching with proxy data (not shown). The spread
in representations of the AABW in PMIP models is thus likely to generate noise in the emergent constraint relationship, but
that noise is either small or compensated.

    We emphasize that this analysis is mainly limited to PMIP2 and PMIP3 models with limited data availability for PMIP4
models, and Southern ocean dynamical feedbacks, which include heat transport and ocean stratification, are suspected of being
model-dependent (Zhu and Poulsen, 2021). Recently, Gregory et al. (In prep.) found a relationship between ECS and AMOC
strength in climate models at pre-industrial. This supports the idea that the lack of correlation between LGM AMOC and
modelled temperatures is influenced by the boundary conditions of the LGM. However, improving SST biases in pre-industrial
oceans at high latitudes is also shown to help LGM modelled AMOC to match proxy data (Sherriff-Tadano and Klockmann,
2021). This might indicate either a limited noise arising from the representations of the AMOC, as biases are replicated between
pre-industrial and LGM, or a larger noise contribution if these biases are enhanced or dampened in the cold LGM with respect
to the warmer 4xCO2 scenario from which ECS is diagnosed. Understanding the full extent of the contribution of the ocean as
a source of noise would require further sensitivity experiments. Notably, variations in ice sheet topography would indicate its
impact on atmosphere and ocean dynamics, or slab ocean simulations would provide information on slow ocean contributions
to LGM temperatures (Zhu and Poulsen, 2021). All in all, the amplitude of the contribution of ocean dynamics in the emergent
constraint relationship is in appearance weak but is potentially underestimated, and it is likely to arise from both LGM boundary
conditions and a decoupling between the cold LGM and the warm abrupt4xCO2.

### 4.6   Cloud feedbacks

Cloud feedbacks ($\lambda_{cl}$) differ substantially across models and contribute to higher ECS in CMIP6 models (e.g. Zelinka et al.,
2020). Because of different boundary conditions, $\lambda_{cl}$ are thought to be regionally different between the LGM and pre-industrial
simulations. We suspect that $\lambda_{cl}$ are state-dependent, i.e. $\lambda_{cl}$ calculated in a cold climate contrasts from the one computed in
abrupt4xCO2 experiments. In this section, we investigate the $\lambda_{cl}$ between cold and warm states for a few climate models, then





**Table 5.** Global and tropical (30° S - 30° N) cloud feedbacks $\lambda_{cl}$ values computed with PRP method for three simulations performed with MPI-ESM1.2-LR.

|  | Global $\lambda_{cl}$ | Tropical $\lambda_{cl}$ |
|---|---|---|
| abrupt2xCO2 | 0.14 | 0.41 |
| LGM abrupt2xCO2 | 0.11 | 0.37 |
| abrupt0p5xCO2 | -0.0 | 0.45 |

analyse the cloud radiative effect (CRE) of several CMIP6 models as a proxy for $\lambda_{cl}$. Finally, we explore the effect of the mixed-phase cloud feedback on the LGM, as it is challenging to represent in models and thought to be one of the main drivers behind the increase of $\lambda_{cl}$ in CMIP6 models.

### 4.6.1 Single-model cloud feedbacks

We calculate $\lambda_{cl}$ in three sets of simulations using MPI-ESM1.2-LR to explore the state-dependency between the LGM and an abrupt2xCO2 experiment. We perform abrupt2xCO2 from pre-industrial $CO_2$, abrupt2xCO2 from LGM $CO_2$ and boundary conditions, and abrupt0p5xCO2 from pre-industrial $CO_2$. Global maps are shown in Fig. 9, and global and tropical calculations of $\lambda_{cl}$ are summarized in Table 5.

In the two $CO_2$ doubling experiments from LGM and pre-industrial states, global $\lambda_{cl}$ are broadly similar, and in contrast
lower when $CO_2$ is halved. This indicates a weak global state-dependency between LGM and pre-industrial, but more pronounced between halving and doubling of $CO_2$. The LGM abrupt2xCO2 notably differs from the pre-industrial abrupt2xCO2 by a $\lambda_{cl}$ near 0 $Wm^{-2}$ over the Laurentide ice sheet. A near-zero $\lambda_{cl}$ above an ice surface is physically plausible since the presence or not of clouds cannot substantially alter the reflection to space. Changes also occur in the Pacific Ocean, where more positive $\lambda_{cl}$ are found in the east, and more negative $\lambda_{cl}$ in the west in the LGM abrupt2xCO2.

State-dependency and forcing dependency of $\lambda_{cl}$ have been assessed in previous studies and models. A slab-ocean version of MIROC3.2 revealed substantially weaker $\lambda_{cl}$ at the LGM than in abrupt2xCO2 experiments (Yoshimori et al., 2009), and similar observations have been made by Zhu and Poulsen (2021) in CESM1.2. However, when comparing halving and doubling of CO2 from pre-industrial conditions, either little differences of global $\lambda_{cl}$ were found in the Australian BRMC model (e.g. Colman and McAvaney, 2009), or weaker $\lambda_{cl}$ in abrupt0p5xCO2 experiment than abrupt2xCO2 experiment in
CESM1.2 (Chalmers et al., 2022).

Generalizing the results of weaker $\lambda_{cl}$ in cold climates of CESM1.2, CESM2.1, MIROC3.2 and MPI-ESM1.2-LR is tempting, but the results from the BRMC model might indicate that the $\lambda_{cl}$ dependency over reduced $CO_2$ compared to increased $CO_2$ could be model-dependent. Moreover, disentangling the dependency of $\lambda_{cl}$ over the ice sheets and the greenhouse gases of the LGM is difficult, and compensations might happen at the global scale, which leads to broadly similar $\lambda_{cl}$ between the LGM





and abrupt2xCO2 experiments. All in all, it is plausible that the influence of $\lambda_{\mathrm{cl}}$ on LGM temperature is model-dependent, and that the decoupling between LGM $\lambda_{\mathrm{cl}}$ and abrupt4xCO2 $\lambda_{\mathrm{cl}}$ is substantial. Since large differences are seen in the tropical Pacific in MPI-ESM1.2-LR and CESM1.2, cloud feedbacks might have been large contributors of the weakness of the tropical emergent constraint.

### 4.6.2 Cross-ensemble variations in CRE change

We assess how large the inter-model spread in $\lambda_{\mathrm{cl}}$ differences between warming and cooling could be among PMIP models based on evidence from single model analyses. To this end, we analyse the cloud radiative effect (CRE) of the CMIP6 models performing abrupt0p5xCO2 and abrupt4xCO2 (Fig. 10). The regression of the change in CRE over surface temperature $\frac{\Delta CRE}{\Delta TS}$ is not the same as $\lambda_{\mathrm{cl}}$, but variations among models in this quantity provide a good estimate of the spread of cloud feedbacks (e.g. Soden et al., 2008). If we suspect a state-dependency on $\lambda_{\mathrm{cl}}$, then the relationships between $\lambda_{\mathrm{cl}}$ and $\frac{\Delta CRE}{\Delta TS}$

should differ whether $\frac{\Delta CRE}{\Delta TS}$ is computed from abrupt0p5xCO2 or abrupt4xCO2.

For almost all models, $\frac{\Delta CRE}{\Delta TS}$ is smaller in abrupt0p5xCO2 than in abrupt4xCO2, which is consistent with our single-model results explored in Section 4.6.1. In Fig. 11, we compare the values of $\frac{\Delta CRE}{\Delta TS}$ for abrupt0p5xCO2 and abrupt4xCO2 with the $\lambda_{\mathrm{cl}}$ of Zelinka et al. (2020) calculated from abrupt4xCO2 simulations. While the intercepts of the two regression lines are broadly similar, the slopes differ, where abrupt4xCO2 is the steepest and abrupt0p5xCO2 the least steep. The difference is more

pronounced at higher $\lambda_{\mathrm{cl}}$ where the $\frac{\Delta CRE}{\Delta TS}$ differ the most. Our analysis indicates a state-dependency in $\lambda_{\mathrm{cl}}$ in CMIP6 models, as the slope across abrupt4xCO2 and abrupt0p5xCO2 ensembles differ in Fig. 11. The state-dependency becomes increasingly stronger with increasing $\lambda_{\mathrm{cl}}$ (i.e. ECS), and as the slope within the abrupt0p5xCO2 ensemble is less steep than abrupt4xCO2, then models can be suspected of being too warm in cooling simulation with respect to their ECS. Lastly, the dispersion of models around the regression lines indicates an inter-model spread in state-dependency on $\lambda_{\mathrm{cl}}$. Since $\lambda_{\mathrm{cl}}$ are calculated from

abrupt4xCO2, the dispersion is minimal around that line, but it is quantified as a standard error of the regression of 0.12 $\mathrm{Wm}^{-2}\mathrm{K}^{-1}$ for abrupt0p5xCO2.

We explore how a variation of 0.12 $\mathrm{Wm}^{-2}\mathrm{K}^{-1}$ of $\lambda_{\mathrm{cl}}$ impacts LGM temperatures by using a low and high ECS model that contributed to both LGM and abrupt0p5xCO2 simulations. In the analysis of Zelinka et al. (2020), MIROC-ES2L and CESM2.1 have ECS of 2.66 K and 5.15 K, respectively. We also use the effective radiative forcing (ERF) of 2xCO2 and

climate feedbacks as calculated in Zelinka et al. (2020). The temperature change can be computed as Eq. 4, and we estimate the LGM forcing for each model as $F_{\mathrm{LGM}} = (\Delta T_{\mathrm{LGM}}/ECS) \times F_{2\mathrm{x}}$, where $\Delta T_{\mathrm{LGM}}$ is -4.05 K for MIROC-ES2L (this study) and -11.3 K for CESM2.1 (Zhu et al., 2021).

$$\Delta T_{\mathrm{LGM}} = \frac{-F_{\mathrm{LGM}}}{\lambda \pm 0.12} \tag{4}$$

For MIROC-ES2L, with an ECS of 2.66 K, ERF at 2xCO2 of 4.11 $\mathrm{Wm}^{-2}$ and a total climate feedback of -1.54 $\mathrm{Wm}^{-2}\mathrm{K}^{-1}$,

the calculated LGM temperature change is within the range -3.8 K to -4.4 K. For CESM2.1, with an ECS of 5.15 K, ERF at 2xCO2 of 3.26 $\mathrm{Wm}^{-2}$ a total climate feedback of -0.63 $\mathrm{Wm}^{-2}\mathrm{K}^{-1}$, the calculated LGM temperature change is within the





range -9.5 K to -14.0 K. Thus, CESM2.1 is more than seven times more sensitive than MIROC-ES2L for a same forcing. This example suggests a larger impact on high ECS models when considering the $\lambda_{cl}$ of abrupt4xCO2 in cooling simulations, such that at the high end ECS, we might see more diversified LGM cooling if more models than CESM2.1 were to run the

simulation. While our analysis are based on the comparison of abrupt4xCO2 and abrupt0p5xCO2, it is reasonable to think that the LGM could also be affected and that models underestimate the cooling. Further analyses on $\lambda_{cl}$ at the LGM are needed, but they indicate that $\lambda_{cl}$ could be a substantial source of noise in the emergent constraint between ECS and LGM temperature.

### 4.6.3 Mixed-phase clouds

The representation of mixed-phase clouds in models are important for ECS (e.g. Gregory and Morris, 1996; Tan and Storelvmo,

2016; Lohmann and Neubauer, 2018). Mixed-phase clouds are notoriously difficult to represent in numerical weather prediction and climate models (Korolev et al., 2017). However, as theory, observations and understanding of mixed-phase cloud processes have improved, their representation in models has changed substantially. Mixed-phase clouds contain a mixture of ice and liquid and exist at temperatures between 0 and -38°C. They have a strong influence on the Earth's energy budget, and the radiative and thermodynamic properties of these clouds depend on the partitioning of cloud liquid water and cloud ice. Liquid clouds

are more reflective than ice clouds and have a longer lifetime, so as the atmosphere warms and cloud ice converts to liquid, cloud albedo and lifetime increase resulting in a negative cloud-phase feedback. The cloud-phase feedback is affected by the mean state of the cloud, and cloud ice processes are important for cloud water phase in GCMs (Komurcu et al., 2014). The representation of mixed-phase clouds in GCMs has changed substantially in models in the past decade, and has been identified as a plausible explanation for the large spread in ECS in CMIP6 models (Zelinka et al., 2020). In this section, we consider how

these changes may have impacted the ECS and the emergent constraint relationship in PMIP.

In PMIP2 models, the physics of mixed-phase clouds were mainly prognostic, with cloud phase dependent on a simple temperature threshold, i.e. liquid turning to ice at -15°C (Sundqvist et al., 1989). In these models the strength of the phase feedback is sensitive to the threshold temperature selected (Gregory and Morris, 1996). As liquid water path is known to exist down to -40°C, the mean-state liquid water path was usually too low and the mean state too icy. This would contribute

to a strong negative cloud-phase feedback, which could lead to lower ECS estimates in these models. By CMIP5/PMIP3, the majority of models implemented ice-nucleation and growth processes in their parameterizations, but cloud liquid is still underestimated at very low temperatures (< -25°C) (Komurcu et al., 2014; Cesana et al., 2015) plausibly contributing to strong cloud-phase phase feedbacks and lower ECS estimates. In CMIP6/PMIP4, cloud water in the mean-state increased in many models and is associated with a weakening of the cloud-phase feedback and an increase in ECS (Zelinka et al., 2020).

Some conjectures are required when considering clouds in past climates as there are no proxy records available and the variables required to analyze clouds are rarely available in the PMIP ensemble. We can speculate that in the colder LGM atmosphere, the low mean state of liquid in mixed-phase clouds in PMIP2 and PMIP3 leads to an overestimate of LGM cooling with respect to the ECS of models, whereas underestimated for PMIP4 models. Finally, the indirect-effect of mineral dust on mixed-phase clouds has not been quantified in any generation of LGM model, although it was predicted to lead to additional

cooling (Sagoo and Storelvmo, 2017). In summary it is likely that the behaviour of mixed-phase clouds under LGM forcing



**Table 6.** Global and tropical (30° S - 30° N) water vapour feedback $\lambda_{wv}$ values computed with PRP method for three simulations performed with MPI-ESM1.2-LR.

|  | Global $\lambda_{wv}$ | Tropical $\lambda_{wv}$ |
| --- | --- | --- |
| abrupt2xCO2 | 2.06 | 2.99 |
| LGM abrupt2xCO2 | 2.07 | 3.03 |
| abrupt0p5xCO2 | 1.84 | 2.61 |

is incomplete, and due to the lack of data, it is challenging to estimate how much noise they could contribute in emergent constraint analysis.

### 4.7 Water vapour feedback

We calculate the water vapor feedback ($\lambda_{wv}$) which is thought to strengthen with warming (e.g. Colman and McAvaney, 2009;
Mauritsen et al., 2019). In Table 6, we report tropical and global $\lambda_{wv}$ in the abrupt2xCO2 experiments, starting from both LGM and pre-industrial, and the abrupt0p5xCO2 with MPI-ESM1.2-LR.

We do not find a large difference in global and tropical $\lambda_{wv}$ in the abrupt2xCO2 experiments, but global $\lambda_{wv}$ is roughly $0.2\ \mathrm{Wm^{-2}}$ lower when halving $CO_2$. A similar observation has been made by Colman and McAvaney (2009) with the BRMC model, but Yoshimori et al. (2009) found no change of $\lambda_{wv}$ between an abrupt2xCO2 and abruptly lowering to LGM greenhouse
gas concentrations, as well as Zhu and Poulsen (2021) in a similar experiment. However, the full LGM simulation revealed lower $\lambda_{wv}$ compared to a warming case (Yoshimori et al., 2009). These results indicate that state-dependency in $\lambda_{wv}$ is also model-dependent, and the inter-model spread might be of the same order of magnitude as that for $\lambda_{cl}$. Although there is an understanding on its increasing strength in warming climates (e.g. Colman and McAvaney, 2009; Mauritsen et al., 2019), it appears that the LGM case introduces additional complexity that may offset this general behavior. We conclude that there is
a possibility of similar to stronger $\lambda_{wv}$ in 4xCO2 experiments compared to the LGM, which could be a source of noise in the emergent constraint.

### 4.8 Methane

Methane emissions from natural sources are intimately linked to global mean temperatures, with decreases in wetland methane emissions during the LGM attributed to a decrease in wetland area and lower rates of methanogenesis due to low CO2 concen-
trations (Valdes et al., 2005). However, most climate models do not treat methane as a feedback, but rather as a forcing since its concentration is prescribed and coupled models rarely have an active chemistry module. In abrupt4xCO2 experiments it is usually kept fixed at the pre-industrial value, whereas in the LGM experiment it is lowered relative to pre-industrial levels.



This makes methane and in general biogeochemical and biophysical feedbacks as systematic biases affecting the emergent constraint between the LGM temperature and the ECS from abrupt4xCO2.

In order to estimate the impact on the LGM emergent constraint of omitting methane feedbacks in abrupt4xCO2 experiments, we perform a simple calculation where we calculate atmospheric methane feedback as $\Delta F_{\mathrm{CH_4}}/\Delta T_{\mathrm{LGM}}$, where $F_{\mathrm{CH_4}}$ is the forcing coming from the decrease of methane at LGM. Methane is well-constrained at the LGM (Loulergue et al., 2008) and is set at 375 ppb in the PMIP4 experiment design (Kageyama et al., 2017). Following the forcing estimates of Etminan et al. (2016), we calculate $\Delta F_{\mathrm{CH_4}} = -0.37$ Wm$^{-2}$ between the LGM and pre-industrial concentrations. With a global temperature

change at LGM of -6.1°C (Tierney et al., 2020), the corresponding methane feedback is 0.06 Wm$^{-2}$K$^{-1}$.

    Recent assessment of all non-CO$_2$ biogeochemical and biogeophysical feedbacks, in which methane feedbacks are included, have a median value of -0.01 Wm-2 (Forster et al., 2021). At the LGM, simulations using WACCM6 (Gettelman et al., 2019) show a 5% colder LGM state than in the CESM2.1 runs, with WACCM6 having a high atmosphere model top and an active chemistry module which better capture the dynamic and chemical changes in ozone, methane and stratospheric water vapour

(Zhu, in prep.). Therefore, even if LGM temperatures are forced by methane feedback, which is not included in the ECS diagnosed from abrupt4xCO2 experiments, fortunately the impact on the emergent constraint relationship is likely to be small.

### 4.9   Paleoclimate SST patterns and their effects

Models have roughly matched reconstructed global mean cooling in the LGM, but have generally problems matching the pattern of cooling with relatively weak temperature change in the tropics and strong polar amplification (e.g. Haywood et al.,

2020; Renoult et al., 2020; Kageyama et al., 2021). In parallel, focusing on the recent historical warming, attention has been paid to how patterns of the SST change can affect feedback mechanisms (e.g. Armour et al., 2013; Ceppi and Gregory, 2017). Current consensus is that the temporary pattern effect in the historical context dampens the rate of warming (Forster et al., 2021). At the same time the long term equilibrium pattern of warming could amplify or dampen the warming (Mauritsen, 2016), but this topic is currently under-explored.

In order for an equilibrium pattern-effect to alter the relationship between ECS and warming or cooling in a given paleoclimate there would have to be a difference in the boundary conditions that would alter the pattern in the paleoclimate in a way that does not happen in the idealised case of 4xCO2. One such example could be if the presence of ice sheets, or differences in the ocean bathymetry or gateways. Yet, if these boundary conditions are included in the respective PMIP case, and models on average respond reasonably to them, then the corresponding pattern-effect is accounted for in the emergent constraint method.

Reconstructed cooling patterns for LGM and warming patterns in the Pliocene are, however, difficult for models to reproduce (e.g. Haywood et al., 2020; Renoult et al., 2020; Kageyama et al., 2021). Foremost, reconstructed climates exhibit a stronger polar amplified response than that simulated by most models. Presumably, such a pattern would activate preferentially the positive high latitude feedbacks at the expense of negative tropical feedbacks, yielding a stronger global mean response which could qualify as an amplifying long term pattern-effect. Yet, the argument requires that models simulate the right long

term pattern to 4xCO2, but not to the LGM and Pliocene boundary conditions.





It would seem that two alternative and simpler explanations are model plausible. First, reconstructions could have exaggerated polar amplifications, for instance due to using different proxies at low and high latitudes or that there are biases in the proxy calibrations that are different at warm and cold climates. Second, models may simply simulate too little polar amplification due to biases in the distribution of for instance cloud feedbacks (Burls and Fedorov, 2014). Since these biases are presumable similar in both 4xCO2 and the Pliocene, it would not affect the emergent constraint relationship, but it would impact the emergent constraint relationship between 4xCO2 and LGM where the biases differ.

## 5   Comparison of the sources of noise

In the previous sections, we have analysed sources of noise which have different geographical and physical impacts over the LGM and therefore contribute to the lack of robustness of the emergent constraint between ECS and LGM temperature. We now discuss the amplitude of each source and the nature of their contribution, which we classify in two categories:

- Structural source: The noise arises from structural uncertainties in the representation of the LGM. The source of noise has an impact on the LGM temperature, but is not expected to influence the ECS of a model calculated from abrupt4xCO2 simulation. The models might simulate lower or higher LGM temperature due to this source of noise, which will mainly modify the median inferred ECS in emergent constraint analysis.

- State-dependent source: The noise exists because of a different behaviour between the LGM and abrupt4xCO2, such that the ECS of models might explain poorly the LGM temperature following the linear relationship given for the emergent constrain framework. Here, we refer to a decoupling between ECS and temperature, where either non-linearity can be seen, or no response at the LGM, which will mainly impact the uncertainty range on inferred ECS.

A summary of the amplitude of each source and whether it is found to be structural or state-dependent is provided in Table 7. Several sources are expected of falling into both categories, in these cases they are classified in both sources of noise.

ice sheet forcing and its efficacy is likely the main source of structural noise as it accounts for half of the radiative forcing at the LGM (Yoshimori et al., 2009; Brady et al., 2013; Kageyama et al., 2017). Ice sheet forcing is highly model-dependent, and studies indicate that the simulated temperature of the LGM might vary to a large extent across models owing to the ice sheet implementation (e.g. Shakun, 2017; Zhu and Poulsen, 2021). So far, studies quantifying this phenomenon are scarce but are valuable as other approaches to calculate ECS from LGM temperature often refer to the ice sheet forcing as an area of high albedo (e.g. Tierney et al., 2020).

The representations of both the AMOC and the Southern ocean dynamics are likely contributors of noise, but their amplitude is less clear. Indeed, we did not find any significant relationship between AMOC and Southern ocean convection and LGM temperature, but their impact might be compensated. The oceans also interact with the sea-ice albedo feedback, and ocean dynamical feedbacks and their connection to sea-ice are found to be important at the LGM in CESM1.2 (Zhu and Poulsen, 2021). The question remains complex, as the structural issues of the AMOC and the Southern ocean might also arises from their pre-industrial representations (Sherriff-Tadano and Klockmann, 2021), implying a state-dependency in the noise. However,





future simulations of the AMOC consistently show a slowdown of the circulation (Weijer et al., 2020; Lee et al., 2021), where in turn, LGM models show slowdown, acceleration or similar to pre-industrial AMOC strength.

The albedo of fixed vegetation, the interaction of snow on vegetation, methane feedback and the drift of model temperature due to restricted spin-up are all small sources of noise. Their impacts are limited, either acting locally (vegetation and snow), or an issue in a few models (vegetation and drift). Vegetation albedo is a structural issue, as in PMIP2 and PMIP3, most models used fixed vegetation maps, which had an impact on LGM temperature independently of their ECS, as a systematic bias. In fact, the true contribution of prescribed vegetation to the LGM cooling is difficult to constrain. It is not thought to be a dominant

factor for the case of the Pliocene warming (Lunt et al., 2012), but that is based on sensitivity experiments which require to interchange ice sheets and vegetation maps between the pre-industrial and the Pliocene. Considering the size of the LGM ice sheets, these sensitivity experiments may be biased and difficult to apply to the LGM. Nevertheless, vegetation ceases to be a structural uncertainty with the implementation of dynamical vegetation feedbacks. It seems difficult to reconcile the weakness of the emergent constraint with differences in vegetation and snow albedo feedbacks, as hypothesized by Hopcroft and Valdes

(2015). However, the additive effect of these small sources may be important in the ensemble.

The contribution of state-dependency in our analysis mainly comes from climate feedbacks. Cloud feedbacks, including mixed-phase cloud feedbacks, are likely to be major drivers of noise in the emergent constraint. Cloud feedbacks, as one of the most uncertain and possibly largest feedbacks, are very model-dependent, with discrepancies in whether they are similar or weaker at LGM compared to a warming climate. They are also found to largely increase the predicted LGM temperature

for higher ECS models. The inter-model spread in water vapour feedback state-dependency is also an important contributor, potentially of similar amplitude as cloud feedbacks. In theory, water vapour feedback is expected of being stronger in a warming climate than at the LGM (e.g. Colman and McAvaney, 2009; Mauritsen et al., 2019), but this is not found in all models, making its behaviour complex. Finally, surface albedo feedback is a smaller source of noise among the climate feedbacks analysed, but likely plays a part in the weakness of the emergent constraint. Here, models mostly agree that surface albedo feedbacks are

amplified in cooling climate, as highly reflective sea-ice and snow-covered areas extend, replacing less reflective biomes such as forests. In fact, the implementation of dynamical vegetation might have amplified the inter-model spread in surface albedo state-dependency, but could have been compensated by the removal of the structural uncertainties arising from prescribed LGM vegetation. As a weaker source of state-dependent noise, we include the physical bound on SST freezing which is, in fact, an issue of decoupling of temperature and abrupt4xCO2 ECS as it mostly arises due to cold pre-industrial SST. However, it is only

a regional issue, and thus is a small contributor of state-dependent noise. The pattern effects are categorized as both structural and state-dependent sources of noise, but the amplitude of their contribution remains difficult to estimate and is the subject of much current research.

All in all, the individual inter-model spread in ice sheet forcing, cloud and water vapour feedback state-dependency might be large enough to significantly change the emergent constraint relationship. However, it is plausible that several sources are

added together, and some others appear as compensated, which could act further in modifying the estimates of ECS from LGM temperature.



Table 7: Assessment of the different sources of noise, as well as whether they arise from structural uncertainties in representing the LGM, or state-dependency and varying strength between cold and warm climate. If the category of noise is written within parenthesis, then it is considered as plausible, but less dominant than the category out of parenthesis.

| Component | Source | Category of noise | Assessment |
|---|---|---|---|
| TOA time series | Temperature drift | Structural | Affects few models. Limited impact. |
| | Energy leakage | State-dependent | Affects few models. Limited impact. |
| Ocean | SST freezing threshold | State-dependent | Affects most models and is regionally critical, but minimal on global scale. |
| | AMOC | Structural or state-dependent) | Large inter-model spread. Could contribute substantially, but likely compensated. |
| | Southern ocean dynamics | Structural or state-dependent) | Large inter-model spread. Could contribute substantially, but likely compensated. |
| Land | Ice sheet forcing | Structural | Large inter-model spread. Limited understanding, but the contribution is expected to be large. |
| | Fixed vegetation | Structural | Systematic bias but limited to land. |
| Surface albedo feedbacks | Snow albedo feedback | State-dependent | Limited source, but could contribute substantially when interacting with sea-ice, ice sheets and vegetation. Likely stronger at the LGM. |
| | Sea-ice albedo feedback | State-dependent | Limited source, but could contribute substantially when interacting with Southern ocean dynamics. Likely stronger at the LGM. |
| | Vegetation albedo feedback | State-dependent | Likely a small contributor. Appeared with the implementation of dynamical vegetation. Likely stronger at the LGM. |





| | | | |
|---|---|---|---|
| Cloud feedbacks | Total cloud feedbacks | State-dependent | Uncertainty direction of sign of state-dependency, from similar to abrupt4xCO2 to weaker. Strongly model-dependent, and is a large contributor of noise. Most likely to affect the tropical Pacific. |
| | Mixed-phase cloud feedback | State-dependent | Responsible for a large spread of ECS among PMIP generations. Currently unclear impact. |
| Others | Water vapour feedback | State-dependent | Uncertain direction of sign of state-dependency, from similar to abrupt4xCO2 to weaker. Could be of similar amplitude as a source of noise as cloud feedbacks. |
| | non-$CO_2$ trace gases | Structural | Omitting methane feedback could lead to a systematic, slight cooling bias. |
| | Pattern effects | Structural or state-dependent | Unclear impact. |

## 6  Statistical view on outlier models and generational issues

A large spread in a model ensemble is advantageous to emergent constraint methods as they are more sensitive to outliers than models near the middle of the ensemble. We explore the influence of CESM2.1, which has an ECS of 5.15 K and cools

3 K more than the next coldest model, on the emergent constraint. We randomly subsample all PMIP models with a simple random sampling approach and display the distributions with and without CESM2.1 in Fig. 12. Excluding CESM2.1 does not impact the median values of slopes and intercepts, which indicates that the behaviour of CESM2.1 is as expected based on the emergent constraint relationship from other models. The correlation coefficient is, however, strongly affected by the inclusion of CESM2.1: when CESM2.1 is sampled, the correlation becomes highly significant.

The robustness of the constraint when CESM2.1 is included is encouraging. Emergent constraint methods do not require a model to be close to the truth, but it should display a physically reasonable relationship between, in our case, the LGM temperature and ECS. In fact having a wide variety of modelled ECS, both low and high, is advantageous. The HadGEM3-GC-31-LL model, which has an ECS of 5.6 K plans to contribute to PMIP4 (Lunt, personal communication), and it will be interesting to see whether its LGM temperature is in line with its high ECS and the emergent constraint relationship. Currently,




with only one high ECS model, the emergent constraint analysis needs to be done with care as the high correlation might be exaggerated when including CESM2.1.

It has been suggested that inherent differences between PMIP generations contribute to the lack of correlation in the emergent constraint. Hopcroft and Valdes (2015) hypothesized that the development of new or more complex model components, for example dynamical vegetation, in PMIP3 relative to PMIP2 would generate additional noise in PMIP3 and be the cause

of the lag of correlation. Thus, this might result in over-confidence in the simpler models used in the emergent constraint of Hargreaves et al. (2012). In Fig. 12, we display the true value of the investigated slope, intercept and correlation coefficient in each PMIP ensemble along with distributions from random sub-samples with and without CESM2.1 described earlier in this section. Despite being relatively close to the lower end, the true value of PMIP2 is always comprised within the 5 – 95% intervals of each parameter analysed in randomly created sub-ensembles without CESM2.1. This indicates that the value of

each parameter for PMIP2 is unlikely to arise from exceptional differences between PMIP2 and other random subensembles. Therefore, the PMIP2 ensemble is not statistically discernible from the other PMIP ensembles and this does not support the hypothesis of Hopcroft and Valdes (2015).

Finally, we inspect the first-generation PMIP1 model ensemble. These models are considerably simpler and lower resolution than more recent models. These models used mixed-layer oceans with prescribed ocean heat transports rather than the three

dimensional dynamic ocean models used from PMIP2 and onwards. It has been suggested that for a given model, using these mixed-layer oceans results in less cooling during the LGM than would arise if a dynamical ocean model component was used (Zhu and Poulsen, 2021). However, from a statistical point of view slope, intercept and correlation of the PMIP1 ensemble is within the ranges of randomly sampled sub ensembles. Therefore, PMIP1 models are not statistically different from later generations in regards to the emergent constraint analysis.

**7   Prospects from single-model ensembles**

Single-model perturbed physics ensembles are a convenient way of avoiding the statistical limitations which may arise from multi-model PMIP sub-ensembles, with restricted ECS and temperature ranges. In most cases, modifying cloud parameters can lead to a wide range of ECS values, but most of these ensembles are made of low complexity models (Gregoire et al., 2011) or slab ocean models (Yoshimori et al., 2011). Slab ocean models are suspected of underestimating the LGM cooling

by missing ocean dynamical feedbacks (Zhu and Poulsen, 2021), and might not be able to perfectly represent the LGM. We analyse two single-model ensembles built out of coupled ESMs, CESM2.1 (Zhu et al., 2022) and MPI-ESM1.2-LR (This study), and plot their temperature and ECS alongside the multi-model ensemble of PMIP in Fig. 13. Simulations have not been run to equilibrium in these ensembles, and the higher ECS versions experience substantial drift of temperature. Thus, interpretations are subject to change.

We estimate a potential equilibrium temperature by regressing the surface temperature time series with the TOA energy imbalance, and show it in Fig. 13. The cooling in the higher ECS versions is amplified, and the regression lines become almost identical to the line in the PMIP4 ensemble. This is an interesting observation, as it could suggest that inter-model structural





uncertainties, which are minimal in single-model ensembles as similar versions share the same ice sheet forcing, contribute in fact little to modifying the emergent constraint relationship. This approach is however limited, as the length of the runs is in

fact too short to reach equilibrium and the control pre-industrial state also has a trend. Therefore, our temperatures obtained via regression might be largely biased. Moreover, we filtered out three versions of CESM2.1 in this analysis, as their estimated temperatures from regression seemed unrealistically low, colder than -20°C and as low as -130°C, which made the slope of the regression line equals to zero.

We create an alternate ensemble using different versions of the CESM model family that are run to equilibrium: CESM1.2,

CESM1.2.3 and CESM2.1. The ECS range of this ensemble is narrower (3.6 – 4.0 K, calculated using abrupt2xCO2 in slab ocean model configuration) and its size smaller (6 models), so we expand the PMIP4 ensemble with it in Fig. 14. CESM2.1 is no longer a strong outlier, with little difference in the constraint whether CESM2.1 is included or removed. Using the tropical SST reconstruction of -3.5°C of Tierney et al. (2020), the median ECS estimate is 3.7 K with CESM2.1 and 3.6 K without. With the tropical SST reconstruction of -2.1°C of Annan et al. (2022), the median ECS is 3.1 K with and without CESM2.1.

These estimates are in broad agreement with approaches based on LGM forcing estimates (e.g. Tierney et al., 2020; Zhu and Poulsen, 2021).

Despite the progress in generating single-model ensembles, they are currently difficult to use in emergent constraint analysis. The ensembles built from CESM2.1 and MPI-ESM1.2-LR are too far from equilibrium and would bias the estimates of ECS. Furthermore, the dependency existing between versions of a same model is also a source of concern for statistical meth-

ods, while still being poorly understood. Recent analyses show that the CESM models are substantially different from each other (Annan et al., 2022), which gives confidence in using them altogether. All in all, single-model ensembles are promising tools to study the sources of noise affecting the emergent constraint, particularly as they are expected to have reduced structural uncertainties and similar ice sheet forcing (e.g. Yoshimori et al., 2011).

## 8 Conclusions

Since its first use in an emergent constraint framework by Crucifix (2006), the LGM temperature has often been considered as among the best paleoclimate candidate to constrain ECS. However, the robustness of the constraint has been substantially decreasing in recent model generations, and the high end of ECS estimates from the LGM are higher than arising from the more uncertain, further in time Pliocene. In this study, we have provided an assessment of the different sources of noise contributing to the weak constraint on ECS from LGM temperature.

– Sources of noise can impact the atmosphere, land and ocean either regionally or globally. Most sources have extra-tropical origins. Inter-model spread in ice sheet forcing and state-dependency in cloud and water vapour feedbacks are considered as the main individual contributors of noise.

– Some sources are associated with structural uncertainties in the simulation of the LGM, they are thus unlikely to affect the ECS of models computed from abrupt4xCO2 experiments. They may bias the LGM temperature (high or low) and





affect the inferred median ECS in emergent constraint analysis. This is the case of ice sheet forcing, ocean dynamics, vegetation albedo, methane feedback, and temperature drift due to limited model spin-up.

- Some sources emerge from the strength of climate components varying between cold and warm climate. These are linked to state-dependency, and leads to a decoupling of the ECS of abrupt4xCO2 and LGM temperature. These sources are likely to affect the uncertainty range of inferred ECS, as they disrupt the linear relationship given in the emergent constraint framework. This is the case of cloud, water vapour and surface albedo feedbacks, as well as the physical bound on SST cooling temperature and SST pattern effects.

- There is no significant difference between the temperatures of PMIP2, PMIP3 and PMIP4. Whereas some sources of noise might be larger in individual PMIPs, such as the state-dependency in surface albedo feedback arising from the implementation of dynamical vegetation, this does not seem to lead to statistically different PMIP ensembles.

- The constraint is critically affected by outlier models, i.e. high and low ECS models. Currently, there is only one high ECS model (CESM2.1, ECS = 5.15 K) which is responsible of the apparent high correlation between LGM temperature and ECS. The inclusion of further high ECS models, such as HadGEM3-GC31-LL, would provide additional information on the high end of the ensemble.

While there are several large sources of noise, the additive impact of smaller sources may be equally important. The quan-
tification of sources remains difficult as it would require sensitivity experiments which do not necessarily exist, and it is likely that compensation processes are at work. Furthermore, the most uncertain sources, such as mixed-phase cloud feedbacks and SST pattern effects, might have been overlooked and could be responsible of a larger noise.

The LGM is currently a weak emergent constraint on ECS, but is broadly consistent with inferences from data-assimilation and forcing estimates methods (e.g. Rohling et al., 2012; Sherwood et al., 2020; Tierney et al., 2020; Zhu and Poulsen, 2021),
but give wider range of uncertainty. This could indicate that either emergent constraint approach are considering too many sources of noise, or that forcing-based methods lead to narrow results.

In comparison, the Pliocene is a robust constraint in both emergent constraint and forcing-based methods (Hargreaves and Annan, 2016; Sherwood et al., 2020; Renoult et al., 2020). This might be due to the state-dependency effect between abrupt4xCO2 and the warm Pliocene being more limited than with the LGM temperature. However, several factors might be
neglected for the case of the Pliocene, notably in structural uncertainties.

The LGM is still the most expanded paleo-ensemble of simulations and therefore remains of particular interest for emergent constraint purposes. Sets of sensitivity experiments involving ice sheet topography, vegetation maps or sea-ice and ocean dynamics would provide useful information on the reasons of noise, as well as the contributions from high ECS models, both which already benefit the Pliocene. This could give a better understanding of poorly understood phenomenons, such a state-
dependent climate feedbacks and ice sheet forcing efficacy, and increase the robustness of the emergent constraint on ECS from LGM temperature.

*Data availability.*  The data of this study are publicly available on the PMIP databases: For PMIP1, http://dods.lsce.ipsl.fr/pmip1db/ (last access: 16 June 2022), PMIP2, http://dods.lsce.ipsl.fr/pmip2_dbext/ (last access: 16 June 2022), PMIP4, http://dods.lsce.ipsl.fr/pmip4/db/ (last access: 16 June 2022). PMIP3 data can be downloaded from the ESGF portal: https://esgf.llnl.gov (last access: 16 June 2022). At the

time of publication, the data of AWI-ESM-1-1-LR, INM-CM4-8, MIROC-ES2L and MPI-ESM1.2-LR are on ESGF. Climate feedbacks data of MPI-ESM1.2-LR can be obtained by asking Martin Renoult. Data of the single-model ensembles of MPI-ESM1.2-LR and CESM2.1 can be asked to Navjit Sagoo and Jiang Zhu, respectively. The data of PlioMIP1 and PlioMIP2 are hosted on a server of University of Leeds. For a username and password, email Alan Haywood (a.m.haywood@leeds.ac.uk) or Julia Tindall (J.C.Tindall@leeds.ac.uk).

*Author contributions.*  The idea of the study is of TM and MR. NS created the single-model ensemble of MPI-ESM1.2-LR runs. JZ provided

the single-model ensemble of CESM2.1 and the multi-model ensemble of CESM model family. JZ provided extended time series of radiation and temperature of CESM2.1. Analyses were performed by MR, with support of NS and JZ for temperatures and radiation of CESM2.1 and MPI-ESM1.2-LR. The paper was written by MR, NS, TM and JZ. TM put the project together.

*Competing interests.*  The authors declare that they have no conflict of interest.

*Acknowledgements.*  We thank Masa Kageyama and Jean-Yves Petterschmidt for the scientific discussions and considerable support in giv-

ing us access to the PMIP1 and PMIP4 model databases. We acknowledge Alan Haywood, Richard Rigby and Julia Tindall for access to the PlioMIP1 and PlioMIP2 databases. We thank all scientists involved in PMIP for sharing data on the PMIP databases and ESGF. We thank Thomas Toniazzo, Kyle Armour, Jennifer Kay, Cristian Proistosescu, Mark Webb, Julia Hargreaves, James Annan, Marie-Luise Kapsch, Sam Sherriff-Tadano, Sandy Harrison, Pascale Braconnot, Dan Lunt, Arthur Oldeman, Michiel Baatsen and Anna von der Heydt for discussions which helped improving this study. The CESM project is supported primarily by the National Science Foundation (NSF). This

material is based upon work supported by the National Center for Atmospheric Research (NCAR), which is a major facility sponsored by the NSF under Cooperative Agreement No. 1852977. Computing and data storage resources, including the Cheyenne supercomputer (doi:10.5065/D6RX99HX), were provided by the Computational and Information Systems Laboratory (CISL) at NCAR. For PMIP outputs, computations and MPI-ESM1.2-LR, the analysis and storage of data were performed on resources provided by the Swedish National Infrastructure for Computing (SNIC) at the National Centre at Linköping University (NSC).



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





**Figure 1.** Emergent constraints for LGM A) tropical and C) global SST anomaly and the ECS of PMIP2, PMIP3 and PMIP4 models. Emergent constraints for Pliocene B) tropical and D) global SST anomaly from PlioMIP1 and PlioMIP2 models. Ordinary least squares regression is calculated, and the coefficients of determination $r^2$ from each sub-ensemble is shown to illustrate the quality of the regression. For the LGM, CESM2.1 is filtered out as discussed in Section 6.



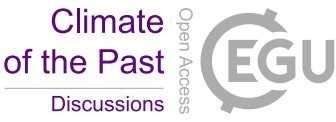

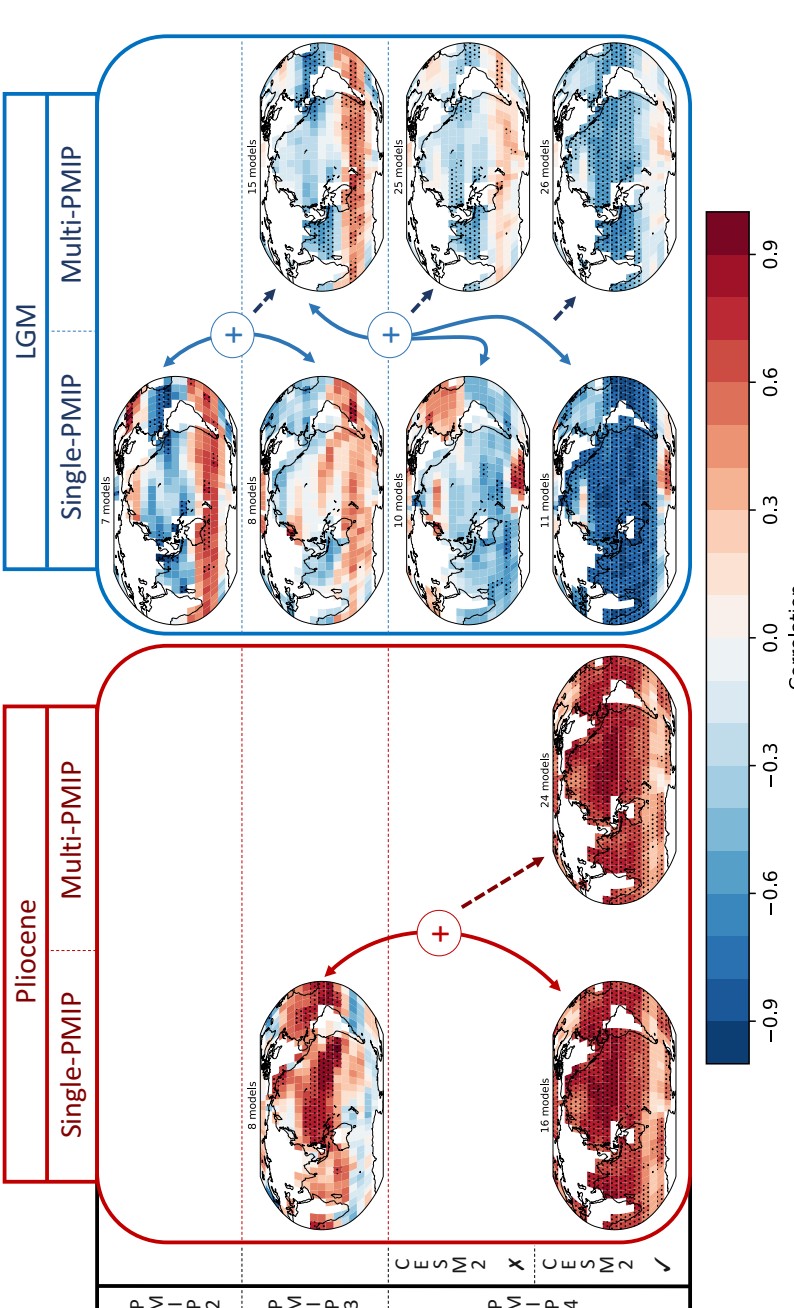

**Figure 2.** Summary of the correlation between SST anomaly and ECS through different PMIP generations. Models have been regridded on 10° grids to minimize dependency between neighboring cells. For both Pliocene and LGM, the left hand side figures are the combination of several generations, following the "+" sign. In PMIP4, the upper row shows ensembles with CESM2.1 included, while lower row shows ensembles excluding CESM2.1. Discussions regarding the presence of CESM2.1 are made in Section 6. Red dotted areas are of positive significance, blue dotted areas are of negative significance under a one-sided t-test (95% threshold).



**Figure 3.** Correlation between SST anomaly and ECS (as in Fig. 2) in A) PMIP2, B) PMIP3, C) PMIP2 + PMIP3 and D) PMIP2 + PMIP3 + PMIP4 (CESM2.1 included) and comparison to a 10-000 member permuted ensemble. If hatched, the correlation in the real ensemble at that cell is outside the 5 - 95% interval of the correlation distribution of the permuted ensemble and is thus unlikely to appear by chance. Models have been regridded on 10° grids to minimize dependency between neighboring cells.





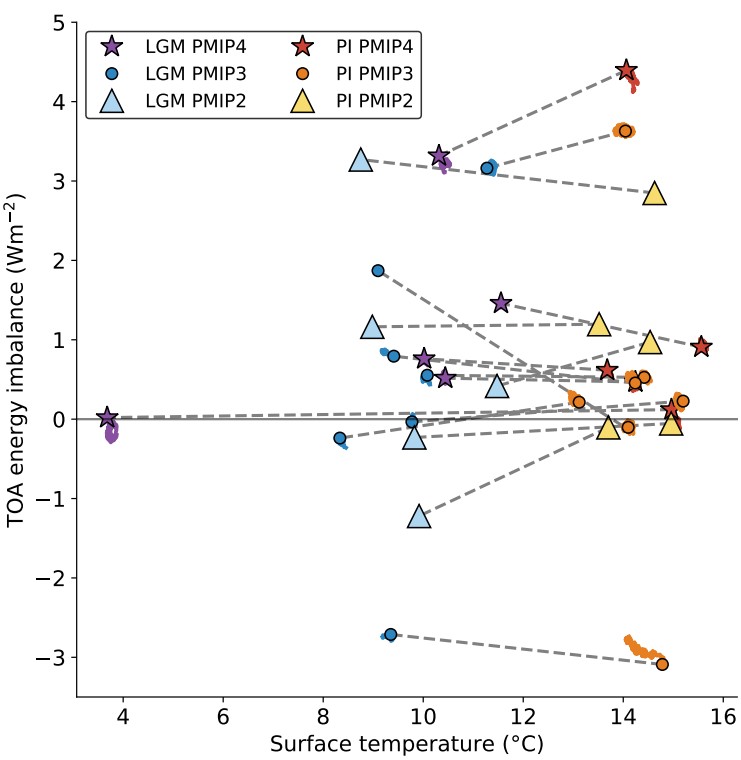

**Figure 4.** Surface temperature (°C) and top-of-atmosphere (TOA) radiation imbalance (Wm$^{-2}$) drift in PMIP models simulating LGM (blue scale) and pre-industrial (PI, orange scale) states. Each trail is a 30-year running mean, while the symbol is a mean of the last 30 years of the time series, when applicable. The gray line connects the LGM and piControl simulation of each model. Note that for PMIP2 models and CCSM4 for PMIP3, time series are not available.



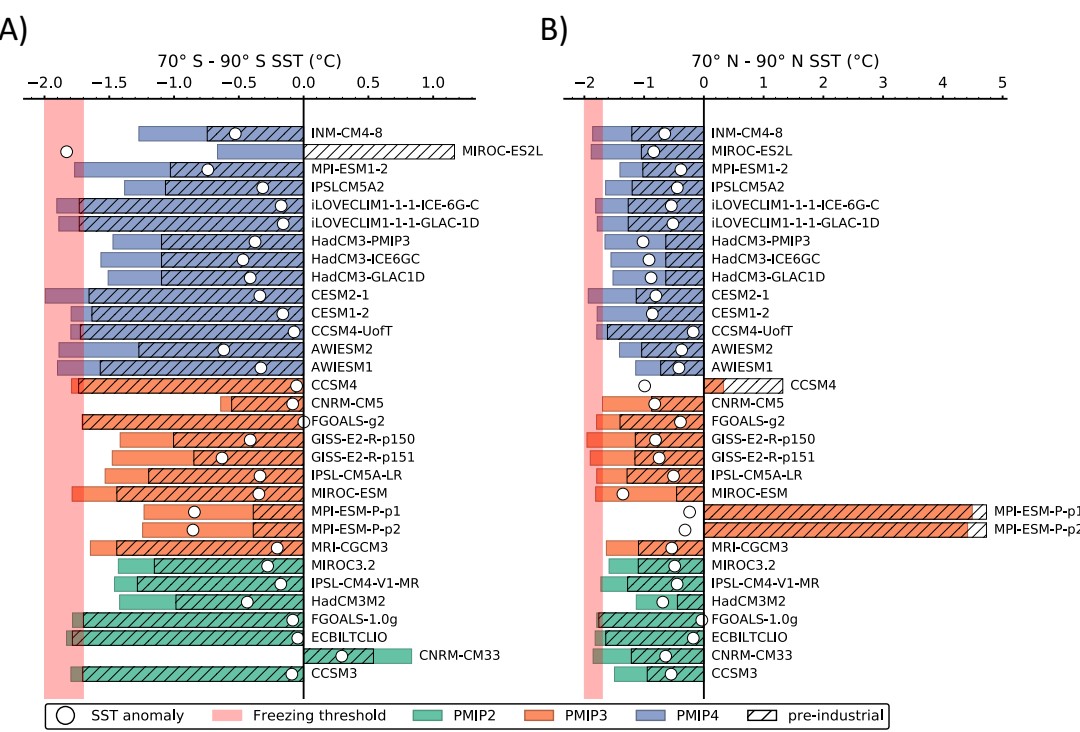

**Figure 5.** SST (°C) of the regions A) south of 70° S and B) north of 70° N in PMIP2, PMIP3 and PMIP4 models in LGM (colored) and pre-industrial (hatched) simulations, as well as SST anomaly between the LGM and pre-industrial (white dot). The red area bounds the -1.7°C – -2.0°C range for freezing point of sea water, which varies among models.



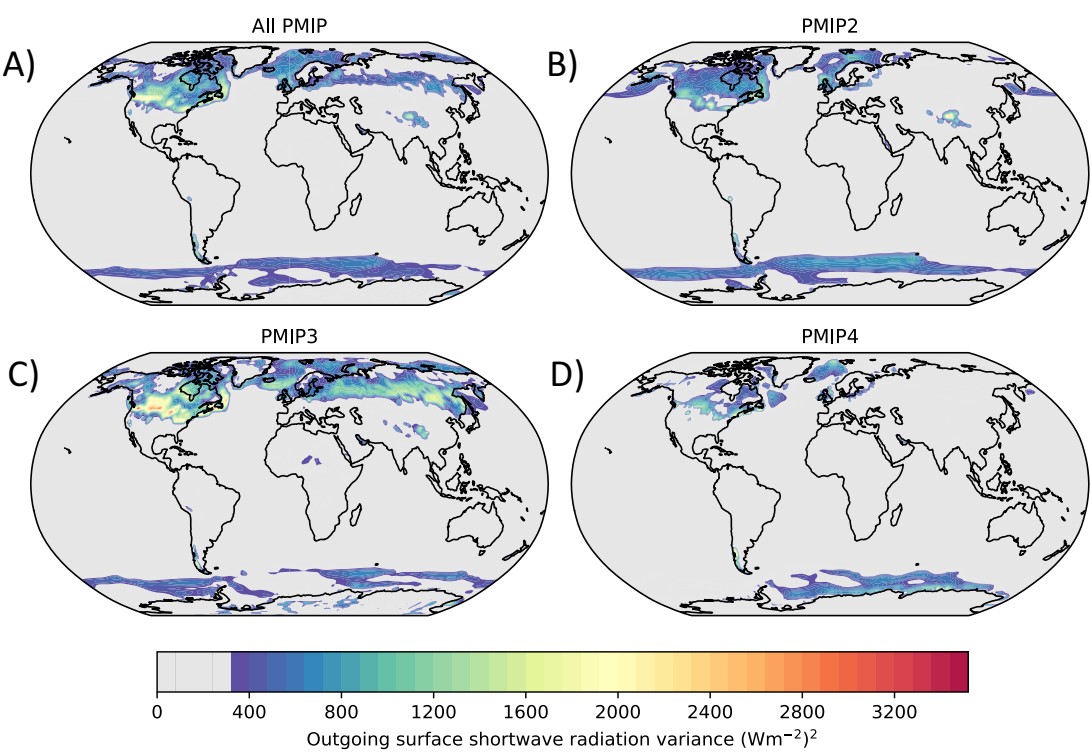

**Figure 6.** Maps of the variance in outgoing (reflected) surface shortwave radiation in the multi-model ensembles of A) all PMIP generations, B) PMIP2, C) PMIP3, D) PMIP4. Variance values which are below less than 10% of the maximum value are masked in grey, to highlight areas of high variance.





**Figure 7.** Maps of the effective surface albedo of B) to F) several PMIP3 models, and comparison with A) the PMIP3 ensemble mean. The ensemble mean excludes the variants GISS-E2-R-p151 and MPI-ESM-P-p2, as they were excluded in the ensembles of Schmidt et al. (2014) and Renoult et al. (2020).





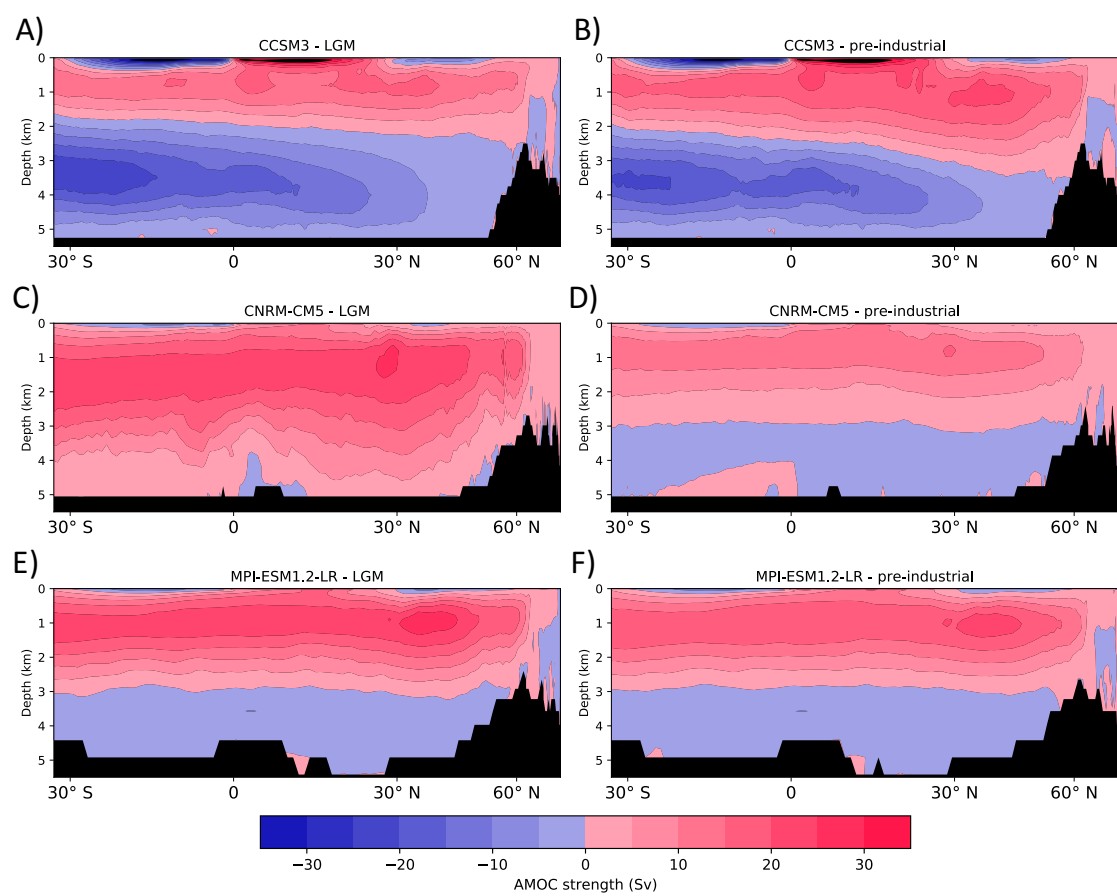

**Figure 8.** Examples of three typical cases of AMOC structure in PMIP models. A) and B), AMOC in agreement with proxy data (CCSM3), C) and D), too deep LGM intrusion of NADW (FGOALS-g2), E) and F) AMOC with little differences between LGM and pre-industrial (MPI-ESM1.2-LR).



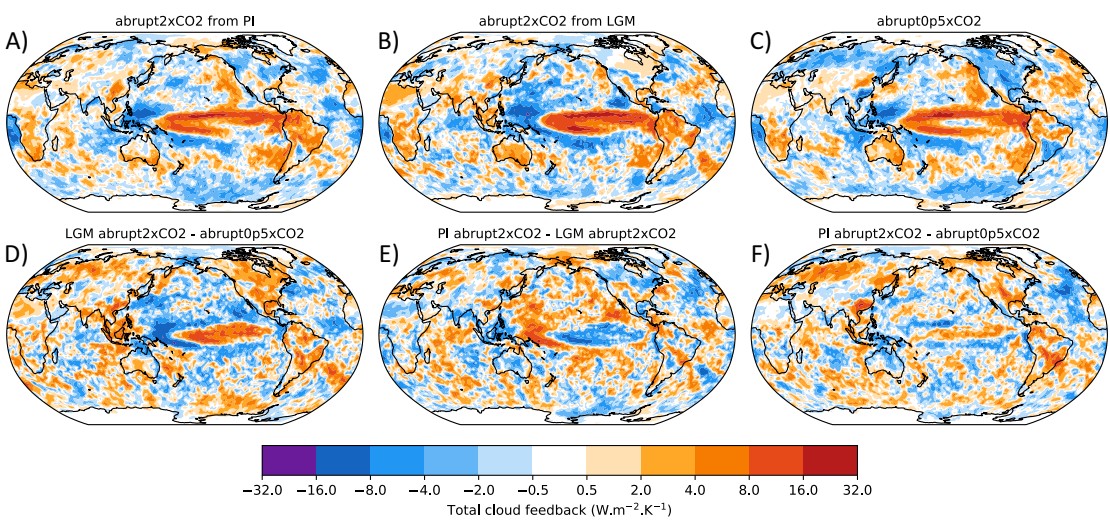

**Figure 9.** Cloud feedback parameters computed with the PRP method in three simulations performed with MPI-ESM1.2-LR (top row) and differences in cloud feedbacks between the three simulations (bottom row). A) abrupt2xCO2 from pre-industrial conditions (560 ppm), B) abrupt2xCO2 from LGM conditions (380 ppm), C) abrupt0p5xCO2 from pre-industrial conditions (180 ppm), D) LGM-abrupt2xCO2 minus abrupt0p5xCO2, E) abrupt2xCO2 minus LGM-abrupt2xCO2 and F) abrupt2xCO2 minus abrupt0p5xCO2.



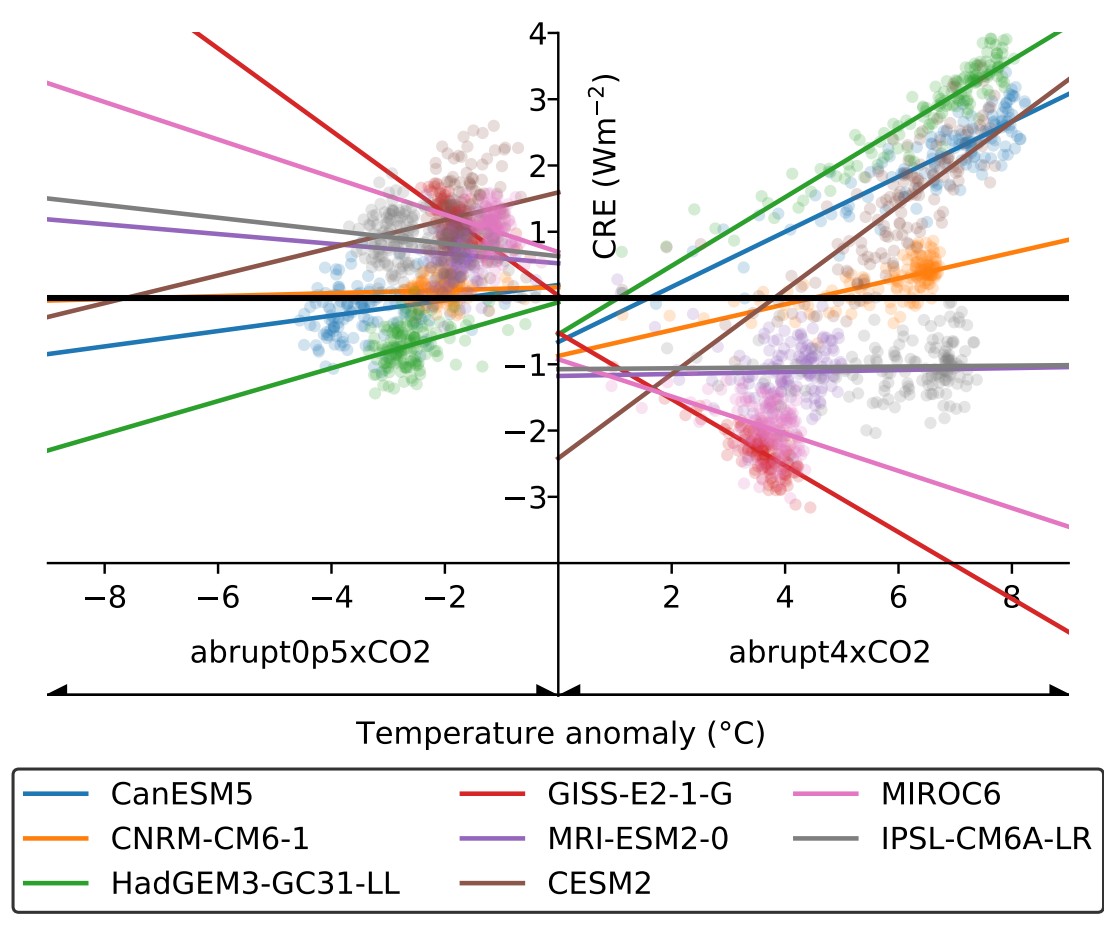

**Figure 10.** Cloud radiative effect (CRE) regressed on global surface temperature anomaly in abrupt0p5xCO2 (left) and abrupt4xCO2 (right) experiments by CMIP6 models.

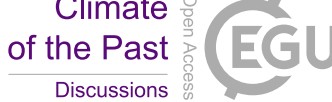

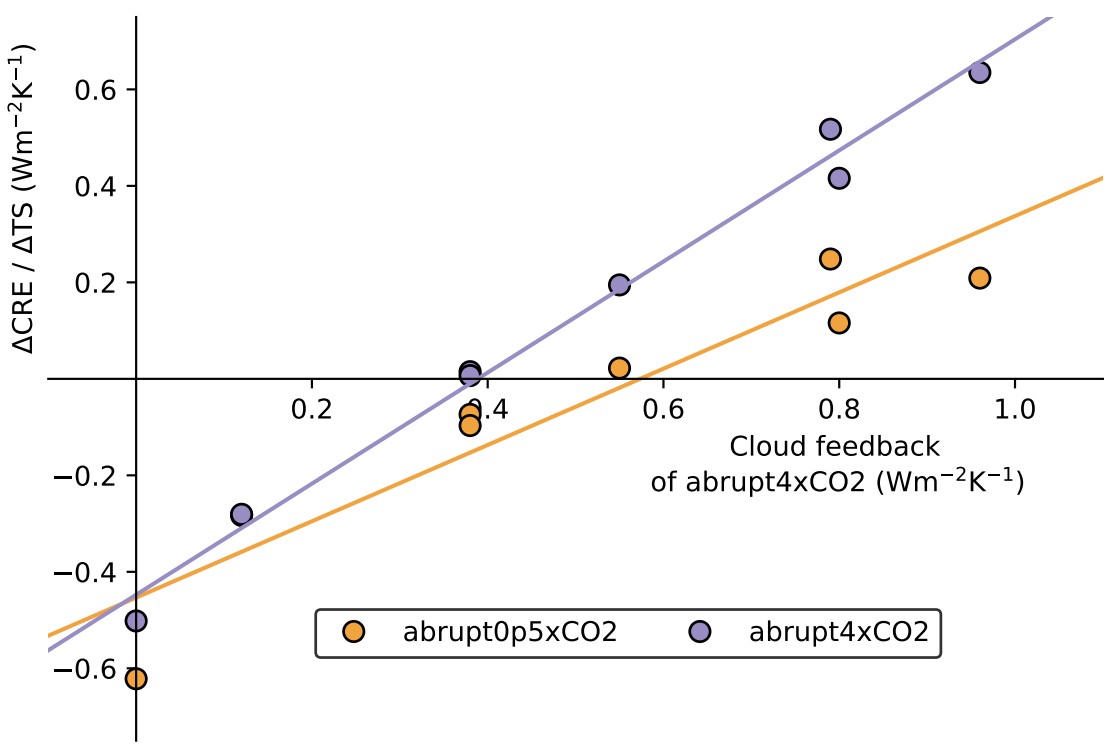

**Figure 11.** Comparison of the $\frac{\Delta CRE}{\Delta TS}$ slopes for abrupt0p5xCO2 and abrupt4xCO2 (shown in Fig. 10) with cloud feedback estimates computed from abrupt4xCO2 in CMIP6 models (values from Zelinka et al. (2020)).





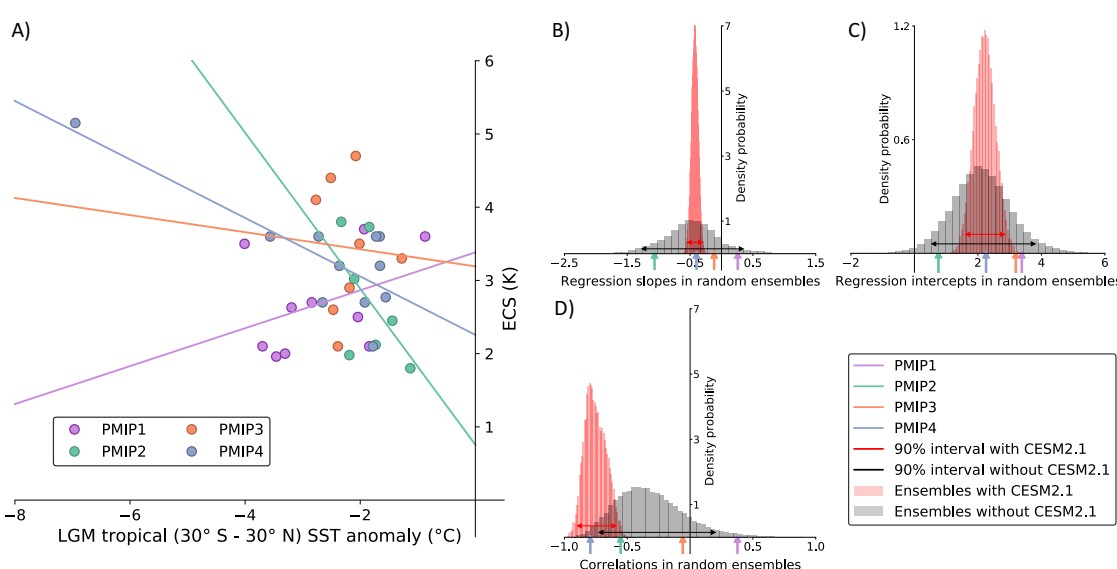

**Figure 12.** A) LGM tropical SST anomaly and ECS of all models since PMIP1, and distribution of regression B) slopes, C) intercepts and D) correlations from randomly sampled sub-ensemble of models. The red distribution shows ensembles where CESM2.1 was sampled, while the black distribution shows ensembles where CESM2.1 was not sampled. The colored arrows pointing at the X axis show the true values of each PMIP ensembles. Note that PMIP1 models are not available for sampling in this analysis, but only shown for comparison with other PMIPs.



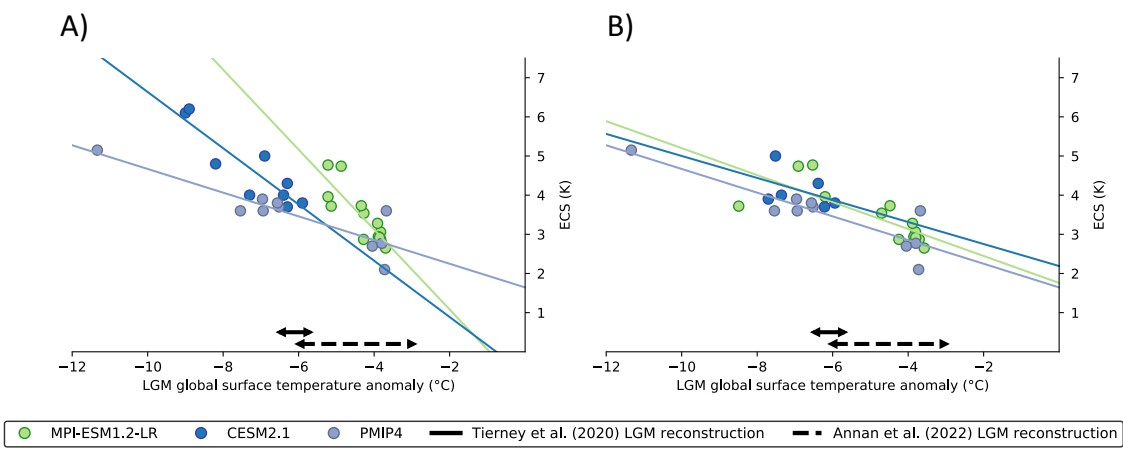

**Figure 13.** Comparison of the emergent constraint relationship between global LGM surface temperature anomaly and ECS in the multi-model ensemble of PMIP4, the single-model ensemble of MPI-ESM1.2-LR (this study) and the single-model ensemble of CESM2.1 (Zhu et al., 2022). A) Surface temperatures are averaged over the last 50 years, B) Surface temperatures are extrapolated by linear regression until top-of-atmosphere imbalance reaches 0 Wm$^{-2}$. The LGM global surface temperature reconstruction of Tierney et al. (2020) is shown, as well as the LGM global surface air temperature reconstruction of Annan et al. (2022).

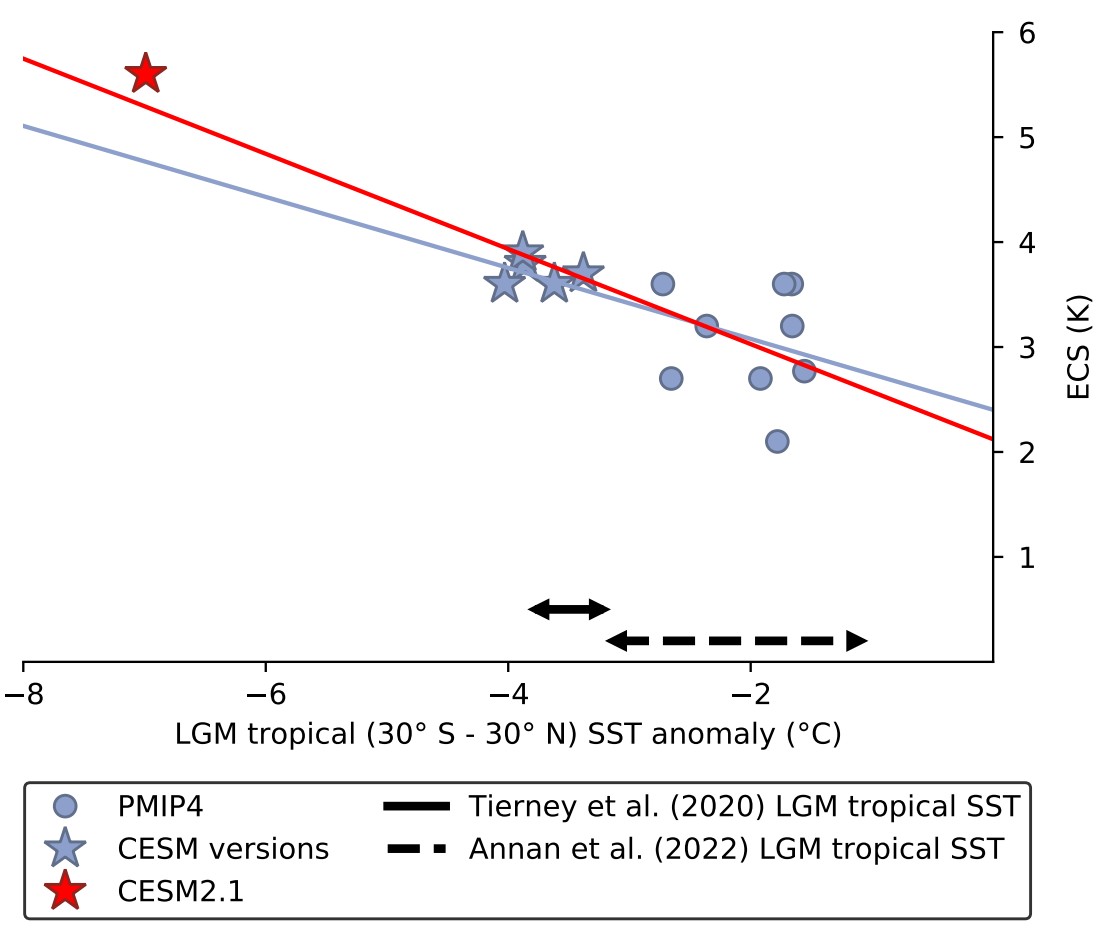

**Figure 14.** Emergent constraint between tropical (30° S - 30° N) LGM SST anomaly and ECS in the PMIP4 ensemble with the addition of several CESM model family versions. The constraint is compared between the presence (red) or absence (blue) of CESM2.1. The ECS of CESM2.1 is 5.6 K, estimated from abrupt2xCO2 in slab ocean mode (Zhu et al., 2021), for comparison with the other CESM versions. The LGM tropical SST reconstructions of Tierney et al. (2020) and Annan et al. (2022) are shown.