# Peer review of "Causes of the weak emergent constraint on climate sensitivity at the Last Glacial Maximum"

_Climate of the Past, 2022_

## Author Response (AR1)

Answer to the editor

We thank the editor for the comments on our manuscript. Here is our answer:

A major issue pointed out is the length of the manuscript, so any possibility of shortening would be welcome.

We believe that substantially shortening the paper would be detrimental to the quality of it. The paper is made of very distinct sections, which are all around 1-page long. A shortening of the paper would be equivalent to removing content; rephrasing some parts of the text would most likely cut only a page or two. Our paper is a thorough assessment of at least 16 years of research on LGM emergent constraint since Crucifix (2006), and 27 years of LGM PMIP simulations since Joussaume and Taylor (1995). Papers of similar lengths are common in the field (Schmidt et al., 2014; Sherwood et al., 2020). These papers are usually well-cited, and are particularly useful for the community and notably the IPCC. Also a large part of the paper is dedicated to references, lengthy in these kind of assessments; our paper is also long due to the number of figures (14), as we made sure to provide as much visuals as possible, which we found more useful to spread this research in the community. The paper got slightly shorter following reviewers comments (around 1-2 pages), and we believe the work on the structure makes it easier for a reader to navigate in the parts of interest.

In contrast, the abstract is quite short so I would recommend also to be a bit more specific.

We modified the abstract to add details and reflect more on the updated content of the paper.

lines 90-95: is this not predictor rather than predictand?

This is indeed a mistake which also appear further down in another line. Corrected.

lines 289-290: CCSM4 and IPSL-CM5A-LR appear to be either far from equilibrium or have substantial leaks and gains of energy, respectively, compared to their pre-industrial states which lie near zero radiation balance (Fig. 4) - but in Figure 4 models are not distinguished, it would be good to put labels to identify them here.

We modified Fig.4 to highlight these models. During the modification of this figure, we realised that it should have been IPSL-CM4 and not IPSL-CM5 as written in the text, so we corrected this typo.

line 312: preindusrial --> preindustrial
- line 512: the mixed-phase cloud feedback might not be obvious to all readers, so please refer to section 4.6.3 here.

Corrected, and added more references to these sections.

Answer to Public Comment of Matthew Huber

We thank Matthew Huber for the encouraging comment on our manuscript and the discussion point risen:

"As a minor point, while I think the discussion of the Pliocene is relevant, some brief discussion of the Eocene, Oligocene, and Miocene might also be relevant. Not as a general, anodyne statement, but as a specific recommendations. If the LGM is a weak candidate for constraining ECS, are there realistic combinations of paleogeographic boundary conditions and climate data/ patterns that would make an excellent candidate? If so, could you speculate a bit about which ones might be better than LGM? The warming signal for example in the Miocene is at least twice as big as in the Pliocene, but with only moderately higher CO2 (https://cp.copernicus.org/articles/ 10/523/2014/ https://agupubs.onlinelibrary.wiley.com/doi/abs/10.1029/2020PA004037) and paleoclimate model

ensembles exist (https://agupubs.onlinelibrary.wiley.com/doi/abs/ 10.1029/2020PA004054). Would that have a higher signal to (non-CO2)-noise ratio? What are the properties of the ideal ECS emergent constraint paleo configuration?"

This comment is an interesting point that we often came across when discussing with other members of the paleoclimate community in conferences. Therefore, we decided to expand the manuscript with the last section entitled "Recommendations on paleo-emergent constraints on ECS", which answers the questions of Matthew Huber. We believe this section fits well with the rest of the manuscript, which, despite being centered on the LGM, also aims to open more on the use of past climates as emergent constraints on ECS.

To summarise this new section, we believe the ideal past climate to be used as emergent constraint on ECS is

1) Warm, with limited ice sheets, considering the substantial noise arising from intermodel differences in ice sheet forcing at the LGM. The climate needs to be warm enough to have a large signal-to-noise ratio, but most likely not as warm as in the Eocene, where non-linearities in feedback might appear.
2) with small changes in paleo-geography due to the issues arising from changes in ocean circulation and how feedbacks behave with varying topographies.
3) with abundant and high-quality proxy data.

We conclude that the Pliocene is likely to be one of the best candidates, but the Miocene is also promising. Oligocene and Eocene are most likely more challenging to use as they are located further in time and therefore have scarce and uncertain data and major changes in paleogeography.

Answer to Anonymous Referee #1

We thank Referee #1 for their careful review of our paper and for the improvements and revisions suggested in their comments. In the following text, we answer to all points discussed by Referee #1, where Referee comments are written as R: and authors comments are written as A:

R: My only comment is concerned with the AMOC: While the authors do classify it as contributing both structural and state-dependent noise (in Table 7 and text), in the discussion the state-dependence becomes a bit vague. I agree that we don't know much, but I think it could be stressed a bit more that it is very likely that the AMOC contributes to state-dependent noise. The AMOC does influence SSTs (and probably globally), but it is not the only factor, and the degree to which the AMOC influences SST can depend on the period of time. See for example the analysis in this preprint: https://doi.org/10.5194/cp-2022-35 for the Pliocene, where an attempt is made to distinguish between AMOC-driven and 'gyre-driven' ocean heat transport. The statement the authors make in line 457-59 may hint towards the fact that for the LGM the amount to which the AMOC influences NH-SSTs is again different from present day and Pliocene.

A: We agree with Reviewer #1 that there should be more emphasis on the state-dependency of the AMOC and how it may influence SST. We have added comments on the state-dependency in ocean heat transport during the Pliocene, as shown by Weiffenbach et al. (2022), and emphasized the complexity of both structural and state-dependent issues in ocean currents as sources of noise in the emergent constraint on ECS. There does not seem to be a clear connection between the AMOC and the warm Pliocene, as AMOC changes seem to have been driven by paleogeography. However, some components of the AMOC may be related to changes in the regional temperatures. We have also referred to this point in the new section on recommendations for future paleo-emergent constraints in response to the public comment of Matthew Huber.  We also use this opportunity to highlight the differences that may arise from simulations being transient (abrupt4xCO2) and near-equilibrium (LGM).

"The state-dependency of the AMOC has been studied at the Pliocene, where models show consistent global SST warming and strengthening of the AMOC compared to pre-industrial (Weiffenbach et al., 2022). Pliocene paleogeography may drive changes in AMOC rather than SSTs (Burton et al., in prep.); however, the warm state of Pliocene north Atlantic SSTs enhance oceanic heat transport by the subtropical gyre which may be responsible for regional SSTs changes (Weiffenbach et al., 2022). An additional caveat is that the simulated LGM is closer to equilibrium response than the 150-year long abrupt4xCO2 simulation from which ECS is diagnosed. There are substantial differences in AMOC strength and structure between transient and equilibrated global warming experiments (Jansen et al., 2018). Generally, decadal time scale transient future simulations of the AMOC show a slowdown of the circulation (Weijer et al., 2020, Lee et al., 2021), where in turn, LGM models show slowdown, acceleration or similar to pre-industrial AMOC strength, suggestive that differences and therefore noise may arise based on how close the simulation is from equilibrium. Overall, this highlights the complexity of structural issues and state-dependencies in ocean currents and how they contribute as sources of noise in the emergent constraint on ECS."

R: Minor comments: line 656: 'model' should be 'more' line 676: capitalize 'ice sheet ...' it is the start of a new sentence. Table 7, row 'Ocean': two brackets too much

A: Corrected.

Answer to Anonymous Referee #2

We thank Referee #2 for pointing out the required clarification and constructive criticism for improvements. In the following text, we answer all the points discussed by Referee #2, where Referee comments are written as R: and authors comments are written as A:.

R: Generally the text reads well, but the flow and structure can be improved. The paper is quite long (the draft has 55 pages!), and the introduction does not place the sections well into context. The different datasets are dropped in without much context. After "1 introduction" comes "2 methodological consideration", "3 regional correlations", "4 Investigation of LGM climate physics", "5 Comparison of the sources of noise" "6 Statistical view on outlier models and generational issues", " 7 Prospects from single-model ensembles" and finally "8 Conclusions", the text meandering along and surprising with nice graphics and well thought-out sections. Please condense structurally and provide more overview in the beginning.

A: We modified the end of the Introduction to provide a better overview of the content of each section. The datasets are described in Methodological considerations, and each subsection of the results has a small introductory paragraph. We believe it is necessary to have this number of section and subsection as our paper covers wide and different topics.

"The paper is organized as the following:
- Section 2: We define the climate sensitivty, temperature variable and emergent constraint theory. We describe the PMIP models and the ensemble of analysis performed to investigate the spread of models.
- Section 3: We extend on methodological considerations by analysing global and regional correlations between temperatures and ECS in the LGM ensembles, as to provide a better view on potential tropical and extratropical biases.
- Section 4: We show the different aspects of the climate system which can be suspected as significant contributors of noise in the emergent constraints. This considers several climate components, i.e. atmosphere, ocean, land surface and cryosphere.
- Section 5: We discuss the results of Section 4, and in particular the contribution and amplitude of noise on the emergent constraint relationship arising from the LGM modelled climate. We categorize the sources of noise as state-dependent or structural.

- Section 6: We further discuss issues of the LGM ensemble which are not be directly connected to the physics of the LGM, such as the effect of outlier models and differences between PMIP generations.
- Section 7: We investigate the current potential of single-model ensembles in emergent constraint on ECS by analysing perturbed physics ensembles of the Max Planck Institute Earth System Model version 1.2 (MPI-ESM1.2-LR), the Community Earth System Model version 2.1 (CESM2.1) and the CESM model family.Section 8: We provide further recommendations on using paleoclimates to constrain ECS. We reflect on the biases affecting the LGM constraint, and evaluate which past climate is ideal for the emergent constraint approach. "

R: Enhance the discussion on limitations. Given that one aim is to "provide a framework for future development of palaeo-emergent constraints" a brief discussion (or at least acknowledgement) of the data/model setup based limitations should be included. One can wonder to what extent ECS is a useful metric in palaeoclimate, given that the system is rarely in equilibrium. The Earth system at beyond-millennial timescales is evolving and feedbacks act across timescales which cannot (yet) be considered with PMIP models. The distinction between "Earth System Sensitivity" and "Climate sensitivity" is not explicitly made, yet it is shown that ice sheet forcing contributes substantially to the radiative forcing and sensitivity.

A: This in an interesting aspect which indeed required further clarifications. In this paper, we apply the definition of ECS as used by the IPCC, which excludes some feedbacks such as the ice sheet feedback. Missing feedbacks are absent from both the abrupt4xCO2 simulation, from which ECS is diagnosed, and the LGM simulation; their inclusion should move models along the already existing regression line, as it represents in some aspect the relationship between temperature / feedbacks between LGM and abrupt4xCO2. Therefore, missing feedbacks which are considered in the definition of ESS are not expected to modify the regression properties. The issues analysed in this study mostly arise from differences between models that are not expected to improve when feedbacks are added. We have added a paragraph to the Discussion section to highlight this point. For the case where the 150-year long abrupt4xCO2 simulation from which ECS is diagnosed is far from equilibrium compared to the LGM simulation, we referred to it in the extended paragraph on the ocean as a sources of noise, as suggested by Reviewer #1.

"It is important to highlight that the definition of ECS is similar to that of the Intergovernmental Panel on Climate Change (IPCC), which includes all feedbacks except the ice sheet feedback. The latter is therefore a missing feedback in both abrupt4xCO2, from which ECS is diagnosed, and in the LGM state. Its inclusion should therefore affect both abrupt4xCO2 and LGM temperatures proportionately, such that climate models would be displaced along the current relationship, and therefore the regression properties should remain similar. The issues analyzed in this study are, for most part, not missing feedbacks, but arise from the lack of consistency between models. These issues are not expected to be reduced with the addition of missing feedback, on the contrary, as models would have more freedom to differ from each other."

R: Pliocene/LGM. Given that the LGM is in the title, once conclusion is that the Pliocene may be a better target to derive emergent constraints. So perhaps the title is not appropriate, and the framing should be adjusted.

A: We believe the title to be appropriate, as the focus of the paper is on the LGM and highlight issues with LGM physics; putting Pliocene in the title may misdirect readers. A paper focused on the Pliocene is planned for submission soon, which should address in details the issues of the Pliocene.

R: - p2 l32: last ice age --> correct to last Glacial period (we are still in an ice age) - correct citation Rohlin et al., 2012 should be PALAEOSENS Project Members. 2012. - Table 7 there are no parentheses (or rather, only two lonely ones)

A: Corrected. We also added a few clarifying words and corrected typos and a reference (Zhu et al., 2022b).